# Rethinking Out-of-Distribution Detection on Imbalanced Data Distribution

**Kai Liu**[1,2*], **Zhihang Fu**[2†], **Sheng Jin**[2], **Chao Chen**[2], **Ze Chen**[2],
**Rongxin Jiang**[1†], **Fan Zhou**[1], **Yaowu Chen**[1], **Jieping Ye**[2]

[1]Zhejiang University,  [2]Alibaba Cloud

## Abstract

Detecting and rejecting unknown out-of-distribution (OOD) samples is critical for deployed neural networks to void unreliable predictions. In real-world scenarios, however, the efficacy of existing OOD detection methods is often impeded by the inherent imbalance of in-distribution (ID) data, which causes significant performance decline. Through statistical observations, we have identified two common challenges faced by different OOD detectors: misidentifying tail class ID samples as OOD, while erroneously predicting OOD samples as head class from ID. To explain this phenomenon, we introduce a generalized statistical framework, termed ImOOD, to formulate the OOD detection problem on imbalanced data distribution. Consequently, the theoretical analysis reveals that there exists a class-aware *bias* item between balanced and imbalanced OOD detection, which contributes to the performance gap. Building upon this finding, we present a unified training-time regularization technique to mitigate the bias and boost imbalanced OOD detectors across architecture designs. Our theoretically grounded method translates into consistent improvements on the representative CIFAR10-LT, CIFAR100-LT, and ImageNet-LT benchmarks against several state-of-the-art OOD detection approaches. Code is available at `https://github.com/alibaba/imood`.

## 1 Introduction

Identifying and rejecting unknown samples during models' deployments, aka OOD detection, has garnered significant attention and witnessed promising advancements in recent years [57, 5, 41, 48, 30]. Nevertheless, most advanced OOD detection methods are designed and evaluated in ideal settings with category-balanced in-distribution (ID) data. However, in practical scenarios, long-tailed class distribution (a typical imbalance problem) not only limits classifiers' capability [7], but also causes a substantial performance decline for OOD detectors [51].

As Wang et al. [51] reveal, a naive combination of long-tailed image cognition [39] and general OOD detection [20] techniques cannot simply mitigate this issue, and several efforts have been applied to study the *joint* imbalanced OOD detection problem [51, 22, 45]. They mainly attribute the performance degradation to misidentifying samples from tail classes as OOD (due to the lack of training data), and concentrate on improving the discriminability for tail classes and out-of-distribution samples [51, 55]. Whereas, we argue that the confusion between tail class and OOD samples presents only one aspect of the imbalance problem arising from the long-tailed data distribution.

To comprehensively understand the imbalance issue, we investigate a wide range of representative OOD detection methods (*i.e.*, OE [20], Energy [32], and PASCL [51]) on the CIFAR10-LT dataset [8]. For each model, we statistic the distribution of wrongly detected ID samples and wrongly detected

---

*Work done during Kai Liu's research internship at Alibaba Cloud. Email: kail@zju.edu.cn.

†Corresponding authors. Email: rongxinj@zju.edu.cn, zhihang.fzh@alibaba-inc.com.

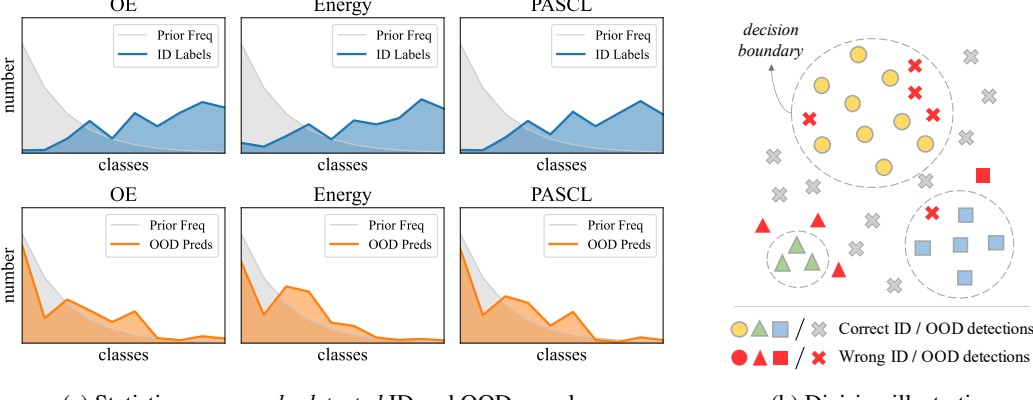

(a) Statistics on *wrongly-detected* ID and OOD samples.     (b) Dicision illustration.

Figure 1: **Issues of OOD detection on imbalanced data**. (a) Statistics of the class labels of ID samples that are *wrongly* detected as OOD, and the class predictions of OOD samples that are *wrongly* detected as ID. (b) Illustration of the OOD detection process in feature space. Head classes' huge decision space and tail classes' small decision space *jointly* damage the OOD detection.

OOD samples, respectively. The results in Fig. 1a reveal that different approaches encounter the same two challenges: (1) ID samples from tail classes are prone to be detected as OOD, and (2) OOD samples are prone to be predicted as ID from head classes. As illustrated in Fig. 1b, we argue that the disparate ID decision spaces on head and tail classes *jointly* result in the performance decline for OOD detection, which has also been confirmed by Miao et al. [40].

To mitigate this problem, Miao et al. [40] developed a heuristic outlier class learning approach (namely COCL) to respectively separate OOD samples from head and tail ID classes in the feature space. Different from COCL, this paper introduces a generalized statistical framework, termed ImOOD, to formulate and explain the fundamental issue of imbalanced OOD detection from a probabilistic perspective. We start by extending closed-set ID classification to open-set scenarios and derive a unified posterior probabilistic model for ID/OOD identification. Consequently, we find that between balanced and imbalanced ID data distributions exists a class-aware *bias* item, which concurrently explains the inferior OOD detection performance on both head and tail classes.

Based on ImOOD, we derive a unified loss function to regularize the posterior ID/OOD probability during training, which simultaneously encourages the separability between tail ID classes and OOD samples, and prevents predicting OOD samples as head ID classes. Furthermore, ImOOD can readily generalize to various OOD detection methods, including OE [20], Energy [32], and BinDisc [5], Mahalanobis-distance [28], *etc.* Besides, our method can easily integrate with other feature-level optimization techniques, including PASCL [51] and COCL [40], to derive stronger OOD detectors. With the support of theoretical analysis, our statistical framework consistently translates into strong empirical performance on the CIFAR10-LT, CIFAR100-LT [8], and ImageNet-LT [51] benchmarks.

Our contribution can be summarized as follows:

- Through statistical observation and theoretical analysis, we reveal that OOD detection approaches collectively suffer from the disparate decision spaces between tail and head classes in the imbalanced data distribution.
- We establish a generalized statistical framework to formulate and explain the imbalanced OOD detection issue, and further provide a unified training regularization to alleviate the problem.
- We achieve superior OOD detection performance on three representative benchmarks, outperforming state-of-the-art methods by a large margin.

## 2    Related Work

**Out-of-distribution detection.** To reduce the overconfidence on unseen OOD samples [4], a surge of post-hoc scoring functions has been devised based on various information, including output

confidence [19, 29, 33], free energy [32, 14, 26], Bayesian inference [36, 9], gradient information [21], model/data sparsity [46, 60, 13, 1], and visual distance [47, 49], *etc*. Vision-language models like CLIP [44] have been recently leveraged to explicitly collect potential OOD labels [17, 16] or conduct zero-shot OOD detections [41]. Other researchers add open-set regularization in the training time [37, 20, 54, 53, 35, 30], making models produce lower confidence or higher energy on OOD data. Manually-collected [20, 52] or synthesized [14, 49] outliers are required for auxiliary constraints.

Works insofar have mostly focused on the ideal setting with balanced data distribution for optimization and evaluation. This paper aims at OOD detection on practically imbalanced data distribution.

**OOD detection on imbalanced data distribution.** In real-world scenarios, the deployed data frequently exhibits long-tailed distribution, and Liu et al. [34] start to study the open-set classification on class-imbalanced setup. Wang et al. [51] systematically investigate the performance degradation for OOD detection on imbalanced data distribution, and develop a partial and asymmetric contrastive learning (PASCL) technique to tackle this problem. Consequently, Wei et al. [55] and [40] extend the feature-space optimization by introducing abstention classes or outlier class learning and integrating with data augmentation or margin learning techniques, respectively. Sapkota and Yu [45] employ adaptive distributively robust optimization (DRO) to quantify the sample uncertainty from imbalanced distributions. Choi et al. [10] focus on the imbalance problem in OOD data, and develop an adaptive regularization for each OOD sample during optimization. In particular, Jiang et al. [23] also utilize class prior to boost imbalanced OOD detector, which is however constrained as a heuristic post-hoc normalization for pre-trained models and unable to translate into training a better detector.

Different from previous efforts, this paper establishes a generalized probabilistic framework to formulate and explain the imbalanced OOD detection issue, and provides a unified training-time regularization technique to alleviate this problem across different OOD detectors.

## 3 Rethinking Imbalanced OOD Detection

In this section, we start from revisiting the closed-set imbalanced image recognition (in-distribution classification), and extend to open-set out-of-distribution detection problem. Finally, we will reveal the class-aware bias between balanced and imbalanced OOD detectors, and derive a unified training-time regularization technique to alleviate the bias for different detectors.

### 3.1 Preliminaries

**Imbalanced Image Recognition**. Let $\mathcal{X}^{in}$ and $\mathcal{Y}^{in} = \{1, 2, \cdots, K\}$ denote the ID feature space and label space with $K$ categories in total. Let $\boldsymbol{x} \in \mathcal{X}^{in}$ and $y \in \mathcal{Y}^{in}$ be the random variables with respect to $\mathcal{X}^{in}$ and $\mathcal{Y}^{in}$. The posterior probability for predicting sample $\boldsymbol{x}$ into class $y$ is given by:

$$P(y|\boldsymbol{x}) = \frac{P(\boldsymbol{x}|y) \cdot P(y)}{P(\boldsymbol{x})} \propto P(\boldsymbol{x}|y) \cdot P(y) \tag{1}$$

Given a learned classifier $f \colon \mathcal{X}^{in} \mapsto \mathbb{R}^K$ that estimates $P(y|\boldsymbol{x})$, in the class-imbalance setting where the label prior $P(y)$ is highly skewed, $f$ is evaluated with the balanced error (BER) [6, 38, 39]:

$$\text{BER}(f) = \frac{1}{K} \sum_y P_{\boldsymbol{x}|y}(y \neq \text{argmax}_{y'} f_{y'}(\boldsymbol{x})) \tag{2}$$

This can be seen as implicitly estimating a class-balanced posterior probability [39]:

$$P^{\text{bal}}(y|\boldsymbol{x}) \propto \frac{1}{K} \cdot P(\boldsymbol{x}|y) \propto \frac{P(y|\boldsymbol{x})}{P(y)} \tag{3}$$

The ideal Bayesian-optimal classification becomes $y^* = \text{argmax}_{y \in [K]} P^{\text{bal}}(y|\boldsymbol{x})$.

**Out-of-distribution Detection**. In the open world, input sample $x$ may also come from the OOD space $\mathcal{X}^{out}$. Let $o$ be the random variable for an unknown label $o \notin \mathcal{Y}^{in}$, and $i$ be the union variable

of all ID class labels (*i.e.*, $i = \cup y$). Given an input $\boldsymbol{x}$ from the union space $\mathcal{X}^{in} \cup \mathcal{X}^{out} \triangleq \mathcal{X}$, the posterior probability for identifying $\boldsymbol{x}$ as in-distribution is formulated as:

$$P(i|\boldsymbol{x}) = \sum_y P(y|\boldsymbol{x}) = 1 - P(o|\boldsymbol{x}) \not\equiv 1 \tag{4}$$

Correspondingly, $P(o|\boldsymbol{x})$ measures the probability that sample $\boldsymbol{x}$ does not belong to any known ID class, aka OOD probability. Hence, the OOD detection task can be viewed as a binary classification problem [5]. Given a learned OOD detector $g\colon \mathcal{X}^{in} \cup \mathcal{X}^{out} \mapsto \mathbb{R}^1$ that estimate the $P(i|\boldsymbol{x})$, samples with lower scores $g(\boldsymbol{x})$ are detected as OOD and vice versa.

### 3.2 Analysis of OOD Detection on Imbalanced Data Distribution

When ID classification meets OOD detection, slightly different from Eq. (1), the classifier $f$ is actually estimating the posterior class probability for sample $\boldsymbol{x}$ from ID space $\mathcal{X}^{in}$ merely, that is, $P(y|\boldsymbol{x}, i)$ [20, 51]. Considering a sample $\boldsymbol{x}$ from the open space $\boldsymbol{x} \in \mathcal{X}^{in} \cup \mathcal{X}^{out}$, the classification posterior in Eq. (1) is re-formulated as $P(y|\boldsymbol{x}) = P(y|\boldsymbol{x}, i) \cdot P(i|\boldsymbol{x})$, the probability that sample $\boldsymbol{x}$ comes from ID data multiply the probability that sample $\boldsymbol{x}$ belongs to the specific $y$-th ID class. According to Eq. (3), we assume the proportion between balanced classification $P^{bal}(y|\boldsymbol{x})$ and vanilla $P(y|\boldsymbol{x})$ for each class $y$ still holds.

**Lemma 3.1.** *For each ID class $y$ in open-set, there exists a non-negative variable $\gamma_y(\boldsymbol{x})$, so that $P^{bal}(y|\boldsymbol{x}) = \gamma_y(\boldsymbol{x}) \cdot \frac{P(y|\boldsymbol{x})}{P(y)}$, where $\gamma_y(\boldsymbol{x}) = \frac{1}{K} \frac{P^{bal}(\boldsymbol{x}|y)}{P(\boldsymbol{x}|y)} \in (0, \infty)$, $P(y|\boldsymbol{x}), P^{bal}(y|\boldsymbol{x}), P(y) \in [0, 1]$.*

In fact, $\gamma_y(\boldsymbol{x})$ captures the likelihood difference for the same sample $\boldsymbol{x}$ between balanced and imbalanced distributions, and also plays the role in constraining the multiplication results to $(0, 1)$. The proof can be found in Appendix A.1. Using Lemma 3.1, we reveal that there exists a class-aware bias term $\beta(\boldsymbol{x})$ between the OOD detection on balanced and imbalanced data distribution.

**Theorem 3.2.** *According to Lemma 3.1, there exists a bias term $\beta(\boldsymbol{x}) = \sum_y \gamma_y(\boldsymbol{x}) \frac{P(y|\boldsymbol{x}, i)}{P(y)}$ between $P^{bal}(i|\boldsymbol{x})$ and $P(i|\boldsymbol{x})$, i.e., $P^{bal}(i|\boldsymbol{x}) = \beta(\boldsymbol{x}) \cdot P(i|\boldsymbol{x})$.*

*Proof.* Since $P(y|\boldsymbol{x}) = P(y|\boldsymbol{x}, i) \cdot P(i|\boldsymbol{x})$, from Lemma 3.1, $P^{bal}(y|\boldsymbol{x})$ can be further expressed as $P^{bal}(y|\boldsymbol{x}) = \gamma_y(\boldsymbol{x}) \cdot \frac{P(y|\boldsymbol{x}, i)}{P(y)} \cdot P(i|\boldsymbol{x})$. According to Eq. (4), it can be formulated as: $P^{bal}(i|\boldsymbol{x}) = \sum_y P^{bal}(y|\boldsymbol{x}) = \sum_y \gamma_y(\boldsymbol{x}) \frac{P(y|\boldsymbol{x}, i)}{P(y)} P(i|\boldsymbol{x}) \triangleq \beta(\boldsymbol{x}) \cdot P(i|\boldsymbol{x})$, where $\beta(\boldsymbol{x}) = \sum_y \gamma_y(\boldsymbol{x}) \frac{P(y|\boldsymbol{x}, i)}{P(y)}$. $\square$

Based on Theorem 3.2, we conclude that the original OOD posterior $P(i|\boldsymbol{x})$ estimated by the detector $g(\boldsymbol{x})$, with the bias term $\beta(\boldsymbol{x})$, causes the performance gap for OOD detection on balanced and imbalanced data distributions. To understand this further, we will first discuss the scenario under ideal conditions and then extend our analysis to real-world scenarios, attempting to analyze the intrinsic bias of the out-of-distribution problem in both cases.

**On ideal class-balanced distribution**, the data likelihood $P(\boldsymbol{x}|y) = P^{bal}(\boldsymbol{x}|y)$, so that $\gamma_y(\boldsymbol{x}) \equiv \frac{1}{K}$. From Theorem 3.2, $\beta(\boldsymbol{x}) = \sum_y \frac{1}{K} \frac{P(y|\boldsymbol{x}, i)}{P(y)}$. Meanwhile, the class prior $P(y) \equiv \frac{1}{K}$, and the summary of in-distribution classification probabilities equals 1 (*i.e.*, $\sum_y P(y|\boldsymbol{x}, i) = 1$), making the bias item $\beta(\boldsymbol{x}) = \sum_y P(y|\boldsymbol{x}, i) = 1$. Ultimately, Theorem 3.2 indicates $P^{bal}(i|\boldsymbol{x}) = P(i|\boldsymbol{x})$, where the detector $g(\boldsymbol{x})$ exactly models the balanced OOD detection.

**On more challenging class-imbalanced distribution**, the class prior $P(y)$ is a class-specific variable for ID categories, and $\beta(x) = \sum_y \gamma_y(\boldsymbol{x}) \frac{P(y|\boldsymbol{x}, i)}{P(y)} \not\equiv 1$. In previous works[20, 51], the class posterior $P(y|\boldsymbol{x}, i)$ is usually estimated with a softmax function by classifier $f$, and the class prior $P(y)$ adopts the sample frequency for each ID class. $\gamma_y(\boldsymbol{x})$ is under-explored and simply treated as a constant. Under this circumstance, since $\sum_y P(y|\boldsymbol{x}, i) = 1$, the bias item can be viewed as a weighted sum of the reciprocal prior $\frac{1}{P(y)}$. Theorem 3.2 explains how $\beta(x)$ causes the gap between balanced (ideal) ID/OOD probability $P^{bal}(i|\boldsymbol{x})$ and imbalanced (learned) $P(i|\boldsymbol{x})$:

- Given a sample $\boldsymbol{x}$ from an ID tail-class $y_t$ with a small prior $P(y_t)$, when the classification probability $P(y_t|\boldsymbol{x}, i)$ gets higher, the term $\beta(\boldsymbol{x})$ becomes larger. Compared to the original $P(i|\boldsymbol{x})$ (learned by $g$), the calibrated probability $P^{\mathrm{bal}}(i|\boldsymbol{x})$ i.e., $P(i|\boldsymbol{x}) \cdot \beta(\boldsymbol{x})$ is more likely to identify the sample $x$ as in-distribution, rather than OOD.
- Given a sample $\boldsymbol{x}'$ from OOD data, as the classifier $f$ tends to produce a higher head-class probability $P(y_h|\boldsymbol{x}', i)$ and a lower tail-class $P(y_t|\boldsymbol{x}', i)$ [23], the term $\beta(\boldsymbol{x}')$ becomes smaller. Compared to the original $P(i|\boldsymbol{x}')$, the calibrated probability $P^{\mathrm{bal}}(i|\boldsymbol{x}')$ i.e., $P(i|\boldsymbol{x}') \cdot \beta(\boldsymbol{x}')$ is more likely to identify the sample $x'$ as out-of-distribution, rather than ID.

The above analysis is consistent with the statistical behaviors (see Fig. 1) of a vanilla OOD detector $g$. Compared to an ideal balanced detector $g^{\mathrm{bal}}$, $g$ is prone to wrongly detect ID samples from tail class as OOD, and simultaneously wrongly detect OOD samples as head class from ID.

### 3.3 Towards Balanced OOD Detector Learning

To address the identified bias in OOD detection due to class imbalance, we present a unified approach from a statistical perspective. Our goal is to push the learned detector $g$ towards the balanced $g^{\mathrm{bal}}$. We outline the overall formula for estimating the OOD posterior below.

Specially, we use the common practices [39, 23] to estimate the probability distribution $P^{\mathrm{bal}}(i|\boldsymbol{x}) = \sum_y P^{\mathrm{bal}}(y|\boldsymbol{x}) = \sum_y \gamma_y(\boldsymbol{x}) \frac{P(y|\boldsymbol{x},i)}{P(y)} P(i|\boldsymbol{x})$ in Theorem 3.2:

First, for the class prior $P(y)$, we use the label frequency of the training dataset [8, 39], expressed as $P(y) := \frac{n_y}{\sum_{y'} n_{y'}} \triangleq \pi_y$, where $n_y$ refers to the number of instances in class $y$. For the class posterior $P(y|\boldsymbol{x}, i)$, given a learned classifier $f$, the classification probability can be estimated using a *softmax* function [39, 23]: $P(y|\boldsymbol{x}, i) := \frac{e^{f_y(\boldsymbol{x})}}{\sum_{y'} e^{f_{y'}(\boldsymbol{x})}} \triangleq p_{y|\boldsymbol{x},i}$, where $f_y(\boldsymbol{x})$ represents the logit for class $y$. For the OOD posterior $P(i|\boldsymbol{x})$, since OOD detection is a binary classification task [5], the posterior probability for an arbitrary OOD detector $g$ [20, 32, 28] can be estimated using a *sigmoid* function: $P(i|\boldsymbol{x}) := \frac{1}{1+e^{-g(\boldsymbol{x})}}$, where $g(\boldsymbol{x})$ is the ID/OOD logit. Finally, for the class-specific scaling factor $\gamma_y(\boldsymbol{x})$, estimating $\gamma_y(\boldsymbol{x})$ is a sophisticated problem in long-tailed image recognition [39, 25]. To focus on the OOD detection problem, we use a parametric mapping $\gamma_{y;\theta} : \boldsymbol{x} \mapsto (0, \infty)$, where $\theta$ are learnable parameters that are optimized through gradient back-propagation.

This unified approach allows us to systematically estimate and correct for the bias introduced by class imbalance, thereby improving the performance of OOD detection in real-world scenarios. According to Theorem 3.2, the balanced OOD detectors $g^{\mathrm{bal}}$ is modeled as:

$$\sigma(g^{\mathrm{bal}}(\boldsymbol{x})) = \left( \sum_y \gamma_{y;\theta}(\boldsymbol{x}) \frac{p_{y|\boldsymbol{x},i}}{\pi_y} \right) \cdot \sigma(g(\boldsymbol{x})) \triangleq \beta(\boldsymbol{x}) \cdot \sigma(g(\boldsymbol{x})) \tag{5}$$

Substitute the sigmoid function $\sigma(z) = \frac{1}{1+e^{-z}}$ into Eq. (5), we have:

$$g(\boldsymbol{x}) = g^{\mathrm{bal}}(\boldsymbol{x}) - \log \left[ (\beta(\boldsymbol{x}) - 1) \cdot e^{g^{\mathrm{bal}}(\boldsymbol{x})} + \beta(\boldsymbol{x}) \right] \tag{6}$$

The derivation is displayed in Appendix A.3. In order to make the detector $g(\boldsymbol{x})$ directly estimate the balanced OOD detection distribution, we can apply the binary cross-entropy loss on calibrated logits:

$$\mathcal{L}_{\mathrm{ood}} = \mathcal{L}_{\mathrm{BCE}} \left( g(\boldsymbol{x}) - \log \left[ (\beta(\boldsymbol{x}) - 1) \cdot e^{g(\boldsymbol{x})} + \beta(\boldsymbol{x}) \right], t \right) \triangleq \mathcal{L}_{\mathrm{BCE}} \left( g(\boldsymbol{x}) - \Delta(\boldsymbol{x}), t \right) \tag{7}$$

where $t = \mathbf{1}\{\boldsymbol{x} \in \mathcal{X}^{in}\}$ indicates whether the sample $\boldsymbol{x}$ comes from ID or not. For a ID sample from tail classes with $t = 1$, discussed in Sec. 3.2, the bias $\beta$ is larger than samples from head classes. In this situation, the punishment $\Delta$ correspondingly increases, which encourages the detector $g$ to generate a higher score $g(\boldsymbol{x})$ to reduce the loss. On the other hand, for a OOD sample that predicted as a head ID class by classifier $f$, $\beta$ and $\Delta$ become smaller than those predicted as tail ID classes, and $g$ are further forced to reduce the score $g(\boldsymbol{x})$ to decrease the loss, as the label $t$ for OOD samples is 0. In practice, to alleviate the optimization difficulty, the $\Delta(\boldsymbol{x})$ for ID samples (with $t = 1$) is cut

off to be non-negative value, and $\Delta(x\prime)$ for OOD samples (with $t = 0$) is cut off to be non-positive values, ensuring the optimization direct dose not conflict with the vanilla BCE loss function.

Meanwhile, according to Lemma 3.1, we add an extra constraint on $\gamma_{y;\theta}$ to ensure the posterior estimate $P^{\text{bal}}(y|\boldsymbol{x}) = \gamma_y(\boldsymbol{x}) \cdot \frac{p_{y|\boldsymbol{x},i}}{\pi_y} \cdot \sigma(g(\boldsymbol{x}))$ and $P^{\text{bal}}(i|\boldsymbol{x}) = \sum_y P^{\text{bal}}(y|\boldsymbol{x})$ will not exceed 1:

$$\mathcal{L}_\gamma = \max \left\{ 0, \sum_y \gamma_{y;\theta}(\boldsymbol{x}) \cdot \frac{p_{y|\boldsymbol{x},i}}{\pi_y} \cdot \sigma(g(\boldsymbol{x})) - 1 \right\} = \max \left\{ 0, \beta(\boldsymbol{x}) \cdot \sigma(g(\boldsymbol{x})) - 1 \right\} \quad (8)$$

Note that for each class $y$, the term $\frac{p_{y|\boldsymbol{x},i}}{\pi_y} \cdot \sigma(g(\boldsymbol{x})) > 0$ always holds, so that we only have to constrain the summary on all classes $P^{\text{bal}}(i|\boldsymbol{x}) = \sum_y P^{\text{bal}}(y|\boldsymbol{x})$ within 1, as indicated in Eq. (8), and $P^{\text{bal}}(y|\boldsymbol{x})$ for each class $y$ will be constrained as well.

Combining with $\mathcal{L}_{\text{ood}}$ and $\mathcal{L}_\gamma$ for optimization, the learned OOD detector $g^*$ has already estimated the balanced $P^{\text{bal}}(i|\boldsymbol{x})$. We thus predict the ID/OOD probability as usual: $\hat{p}(i|\boldsymbol{x}) = \frac{1}{1+e^{-g^*(\boldsymbol{x})}}$. The other terms like $\gamma$, $\beta$, and $\Delta$ are no longer needed to compute, maintaining the inference efficiency and simplicity for OOD detection applications.

## 4 Experiments

In this section, we empirically validate the effectiveness of our ImOOD on several representative imbalanced OOD detection benchmarks. The experimental setup is described in Sec. 4.1, based on which extensive experiments and discussions are displayed in Sec. 4.2 and Sec. 4.3.

### 4.1 Setup

**Datasets.** Following the literature [51, 23, 10, 40], we use the popular CIFAR10-LT, CIFAR100-LT [8], and ImageNet-LT [34] as imbalanced in-distribution datasets.

For CIFAR10/100-LT benchmarks, the imbalance ratio (*i.e.*, $\rho = \max_y(n_y)/\min_y(n_y)$) is set as 100 [8, 51]. The original CIAFR10/100 test sets are kept for evaluating the ID classification capability. For OOD detection, the TinyImages80M [50] is adopted as the auxiliary OOD training data, and the test set is semantically coherent out-of-distribution (SC-OOD) benchmark [56].

For the large-scale ImageNet-LT benchmark, training samples are sampled from the original ImageNet-1k [12] dataset, and the validation set is taken for evaluation. We follow the OOD detection setting as Wang et al. [51] to use ImageNet-Extra as auxiliary OOD training and ImageNet-1k-OOD for testing. Randomly sampled from ImageNet-22k [12], ImageNet-Extra contains 517,711 images belonging to 500 classes, and ImageNet-1k-OOD consists of 50,000 images from 1,000 classes. All the classes in ImageNet-LT, ImageNet-Extra, and ImageNet-1k-OOD are not overlapped.

**Evaluation Metrics.** For OOD detection, we report three metrics: (1) AUROC, the area under the receiver operating characteristic curve, (2) AUPR, the area under the precision-recall curve, and (3) FPR95, the false positive rate of OOD samples when the true positive rate of ID samples are 95%. For ID classification, we measure the macro accuracy of the classifier. We report the mean and standard deviation of performance (%) over six random runs for each method.

**Implementation Details.** For the ID classifier $f$, following the settings of Wang et al. [51], we train ResNet18 [18] models on the CIFAR10/100LT benchmarks, and leverage ResNet50 models for the ImageNet-LT benchmark. Logit adjustment loss [39] is adopted to alleviate the imbalanced ID classification. Detailed settings are displayed in Appendix B.1. For the OOD detector $g$, as Bitterwolf et al. [5] suggest, we implement $g$ as a binary discriminator (abbreviated as *BinDisc*) to perform ID/OOD identification. Detector $g$ shares the same backbone (feature extractor) as classifier $f$, and $g$ only attaches an additional output node to the classification layer of $f$. In addition, we also add a linear layer on top of the backbone to produce the $\gamma$ factors to perform the training regularization with Eq. (7) and Eq. (8). To reduce the optimization difficulty, the gradient is stopped between Eq. (7) and Eq. (8) (but still shared in the backbone), where Eq. (7) aims at training $g$ while Eq. (8) only optimize $\gamma$. Furthermore, to verify the versatility of our method, we also implement several representative

Table 1: OOD detection evaluation on CIFAR10/100-LT benchmarks. The best results are marked in **bold**, and the secondary results are marked with underlines. The base model is ResNet18.

| Method | CIFAR10-LT | | | | CIFAR100-LT | | | |
|---|---|---|---|---|---|---|---|---|
| | AUROC↑ | AUPR↑ | FPR95↓ | ACC↑ | AUROC↑ | AUPR↑ | FPR95↓ | ACC↑ |
| MSP | 72.28 | 70.27 | 66.07 | 72.34 | 61.00 | 57.54 | 82.01 | 40.97 |
| OECC | 87.28 | 86.29 | 45.24 | 60.16 | 70.38 | 66.87 | 73.15 | 32.93 |
| EnergyOE | 89.31 | 88.92 | 40.88 | 74.68 | 71.10 | 67.23 | 71.78 | 39.05 |
| OE | 89.77 | 87.25 | 34.65 | 73.84 | 72.91 | 67.16 | 68.89 | 39.04 |
| PASCL | 90.99 | 89.24 | 33.36 | 77.08 | 73.32 | 67.18 | 67.44 | 43.10 |
| OpenSampling | 91.94 | 91.08 | 36.92 | 75.78 | 74.37 | 75.80 | 78.18 | 40.87 |
| ClassPrior | 92.08 | 91.17 | 34.42 | 74.33 | 76.03 | 77.31 | 76.43 | 40.77 |
| BalEnergy | 92.56 | 91.41 | 32.83 | 81.37 | 77.75 | 78.61 | 73.10 | 45.88 |
| EAT | 92.87 | 92.40 | 28.83 | 81.31 | 75.45 | 70.87 | **64.01** | 46.23 |
| COCL | 93.28 | 92.24 | 30.88 | 81.56 | 78.25 | 79.37 | 74.09 | 46.41 |
| PASCL + **Ours** | 92.93 | 92.51 | 28.73 | 78.96 | 74.23 | 68.63 | 65.65 | 44.60 |
| COCL + **Ours** | **93.55** | **92.83** | **28.52** | **81.83** | **78.50** | **79.96** | 71.65 | **46.80** |

OOD detection methods (*e.g.*, OE [20], Energy [32], *etc.*) into binary discriminators, and equip them with our ImOOD framework. For more details please refer to Appendix B.2.

**Methods for comparison.** In the following sections, we mainly compare our method on three benchmarks with the typical OOD detectors including OE [20], Energy [32], and BinDisc [5], as well as some state-of-the-art detectors such as PASCL [51], ClassPrior [23], EAT [55], COCL [40], *etc.*. Specifically, as the results on the ImageNet-LT benchmark reported by COCL share a large discrepancy against PASCL (especially the AUPR measure), we re-implement COCL based on their released code[1] and report the aligned results in Tab. 2.

## 4.2 Main Results

**ImOOD significantly outperforms previous SOTA methods on CIFAR10/100-LT benchmarks.** As shown in Tab. 1, our ImOOD achieves new SOTA performance on both of CIFAR10/100-LT benchmarks. Built on top of the strong baseline PASCL [51], our method leads to 1.9% increase of AUROC, 3.2% increase of AUPR, and 4.6% decrease of FPR95 on CIFAR10-LT, with 0.9% - 1.8% enhancements of respective evaluation metrics on CIFAR100-LT. By integrating with COCL [40], our ImOOD further pushes the imbalanced OOD detection on CIFAR10/100-LT benchmarks towards a higher performance, *e.g.*, achieving 93.55%/78.50% of AUROC respectively. To further demonstrate the efficacy, we validate our method on the real-world large-scale ImageNet-LT [34] benchmark, and the results are displayed below.

Table 2: OOD detection evaluation on the ImageNet-LT benchmark. The base model is ResNet50.

| Method | AUROC↑ | AUPR↑ | FPR95↓ | ACC↑ |
|---|---|---|---|---|
| MSP | 53.81 | 51.63 | 90.15 | 39.65 |
| OECC | 63.07 | 63.05 | 86.90 | 38.25 |
| EnergyOE | 64.76 | 64.77 | 87.72 | 38.50 |
| OE | 66.33 | 68.29 | 88.22 | 37.60 |
| PASCL | 68.00 | 70.15 | 87.53 | 45.49 |
| EAT | 69.84 | 69.25 | 87.63 | 46.79 |
| COCL | 73.87 | 72.63 | 76.35 | 51.00 |
| PASCL + **Ours** | 74.69 | 73.08 | 74.37 | 46.63 |
| COCL + **Ours** | **75.84** | **73.19** | **74.96** | **52.43** |

---
[1]https://github.com/mala-lab/COCL

**ImOOD achieves superior performance on the ImageNet-LT benchmark.** As Tab. 2 implies, our ImOOD brings significant improvements against PASCL, *e.g.*, 6.7% increase on AUROC and 13.2% decrease on FPR95, and further enhance the SOTA method COCL for a better OOD detection performance, *e.g.*, 75.84 of AUROC. Since the performance enhancement is much greater than those on CIFAR10/100-LT benchmarks, we further statistic the class-aware error distribution on wrongly detected ID/OOD sample in Fig. A1. The results indicate our method builds a relatively better-balanced OOD detector on ImageNet-LT, which leads to higher performance improvements. Besides, as we follow the literature [51, 40] to employ ResNet18 on CIFAR10/100-LT while adopt ResNet50 on ImageNet-LT, the model capacity also seems to play a vital role in balancing the OOD detection on imbalanced data distribution, particularly in more challenging real-world scenarios.

**Additional comparison following ClassPrior's setting.** Since ClassPrior [23] uses a totally different setting against the literature [51, 40] on the ImageNet datasets, including different ID imbalance ratio and OOD test sets, we additionally compare with ClassPrior in Tab. 3. For a fair comparison, as ClassPrior does not leverage real OOD data for training, we eliminate the auxiliary ImageNet-Extra dataset and utilize the recent VOS [14] technique to generate OOD syntheses for our regularization. According to Tab. 3, our method consistently outperforms ClassPrior by a large margin on all subsets.

Table 3: Comparison on ClassPrior's ImageNet-LT-a8 benchmark. The base model is MobileNet.

| Method | iNaturalist | | SUN | | Places | | Textures | |
|---|---|---|---|---|---|---|---|---|
| | AUROC↑ | FPR95↓ | AUROC↑ | FPR95↓ | AUROC↑ | FPR95↓ | AUROC↑ | FPR95↓ |
| ClassPrior | 82.51 | 66.06 | 80.08 | 69.12 | 74.33 | 79.41 | 69.58 | 78.07 |
| **Ours** | **86.15** | **59.13** | **81.29** | **65.88** | **77.57** | **76.26** | **72.82** | **72.73** |

## 4.3 Ablation Studies

In this section, we conduct in-depth ablation studies on the CIFAR10-LT benchmark to assess the validity and versatility of our proposed ImOOD framework, and the results are reported as follows.

Table 4: Ablation on the $\gamma_y$ estimates and technique integration on the CIFAR10-LT benchmark.

| $\gamma_y$ Estimates | AUROC↑ | AUPR↑ | FPR95↓ | ID ACC↑ |
|---|---|---|---|---|
| none | 90.06 | 88.72 | 33.39 | 78.22 |
| $\gamma_y := const$ | 89.75 | 86.28 | 32.67 | 78.50 |
| $\gamma_y := \gamma_{y;\theta}$ | 92.04 | 91.32 | 31.24 | 79.16 |
| $\gamma_y := \gamma_{y;\theta}(\boldsymbol{x})$ | 92.23 | 91.92 | 29.95 | 79.56 |
| + PASCL | 92.93 | 92.51 | 28.73 | 78.96 |
| + COCL | **93.55** | **92.83** | **28.52** | **81.83** |

**Lemma 3.1 ($P^{\textbf{bal}}(y|\boldsymbol{x}) = \gamma_y(\boldsymbol{x}) \cdot \frac{P(y|\boldsymbol{x})}{P(y)}$) is consistent with empirical results.** To validate that the coefficient $\gamma_y(\boldsymbol{x})$ depends on input sample $\boldsymbol{x}$ and differs for each class $y$, we perform a series of ablation studies in Tab. 4. We first build a baseline model with BinDisc [5] only, and no extra regularization is adopted to mitigate the imbalanced OOD detection. As shown in the first row from Tab. 4, the baseline (denoted as *none* of $\gamma_y$ estimates) presents a fair OOD detection performance (*e.g.*, 90.06% of AUROC and 33.39% of FPR95). Then, we simply take $\gamma_y$ as a constant for all classes (denoted as $\gamma_y := const$) by assuming $P^{\text{bal}}(\boldsymbol{x}|y) = P(\boldsymbol{x}|y)$ and $\gamma_y := \frac{1}{K}$ (see Appendix A.1) to apply the training regularization in Sec. 3.3. According to Tab. 4, the OOD detection performance receives a slight decline of AUROC (from 90.06% to 89.75%), despite the better FPR95 result. Consequently, after treating $\gamma_y$ as a learnable variable for each class $y$ (denoted as $\gamma_y := \gamma_{y;\theta}$), the detector receives significant improvement on all the three measures of AUROC, AUPR, and FPR95. Finally, setting $\gamma_y$ as an input-dependent and class-aware learnable variable (denoted as $\gamma_y := \gamma_{y;\theta}(\boldsymbol{x})$) brings further OOD detection enhancement.

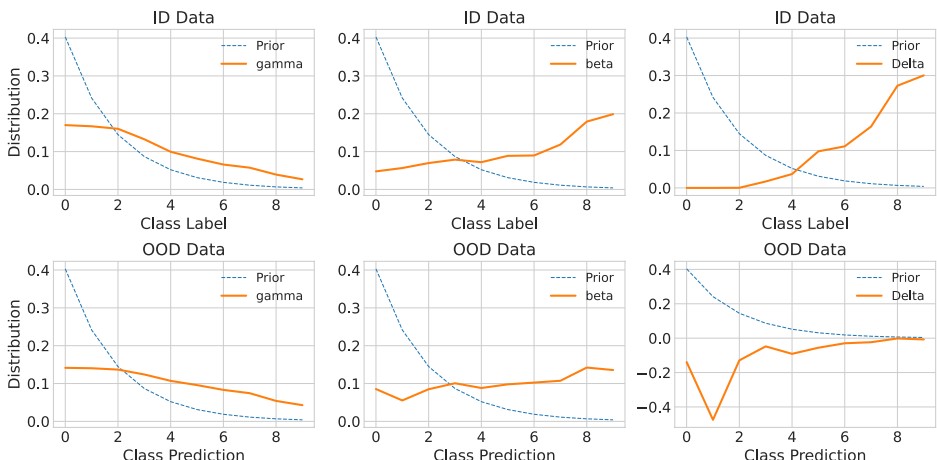

Figure 2: **Statistics on $\gamma$, $\beta$, and $\Delta$ from the CIFAR10-LT benchmark**. (1) Upper: distributions on ID samples from head to tail (left to right) class indices; (2) Lower: distributions on OOD samples predicted as head to tail (left to right) ID classes.

In addition, we further statistics the practical distribution of $\gamma$ in Fig. 2, where $\gamma_y(\boldsymbol{x})$ is relatively higher for the ID sample from head classes. It is consistent with $\gamma_y(\boldsymbol{x}) = \frac{1}{K}\frac{P^{\mathrm{bal}}(\boldsymbol{x}|y)}{P(\boldsymbol{x}|y)}$ (see Appendix A.1), as the data likelihood $P(\boldsymbol{x}|y)$ is close to the balanced situation ($P^{\mathrm{bal}}(\boldsymbol{x}|y)$) for head samples while over-estimated for tail classes, leading to a higher fraction of $\frac{P^{\mathrm{bal}}(\boldsymbol{x}|y)}{P(\boldsymbol{x}|y)}$ for head classes and lower for tail classes. Detailed discussion is presented in Appendix A.2.

**The designed training-time regularization in Sec. 3.3 is effective.** As shown in Tab. 4, compared with the BinDisc baseline in the first row, adding Eq. (7) and Eq. (8) by setting $\gamma_y := \gamma_{y;\theta}(\boldsymbol{x})$ leads to significant improvement on the OOD detection performance (*i.e.*, 2.2% increase of AUROC and 3.4% decrease of FPR95). As Fig. 2 shows, the automatically learned adjustments (*e.g.*, $\beta$ and $\Delta$) are consistent with our motivation, where we aim to punish the ID data from tail classes with a large positive $\Delta$, as well as the OOD samples predicted as ID head classes with a large negative $\Delta$ (Sec. 3.3). Moreover, as indicated by Tab. 4, integrating with PASCL [51] and COCL [40] techniques further boosts the imbalanced OOD detection, ultimately resulting in a new SOTA performance.

**ImOOD generalizes to various OOD detection methods.** To verify the versatility of our statistical framework, we implement the ImOOD framework with different OOD detectors beside BinDisc, including the OE [20], Energy [32], and the Mahalanobis distance [24]. For those detectors, we add an extra linear layer to conduct logistic regression on top of their vanilla OOD scores (see Appendix B.2), and leverage our training-time regularization to optimize those detectors in a unified manner. According to the results presented in Tab. 5, our ImOOD consistently boosts the original OOD detectors by a large margin on all the AUROC, AUPR, and FPR95 measures. In real-world applications, one may choose a proper formulation of our ImOOD to meet specialized needs.

Table 5: Generalization to OOD detectors.

| OOD Detector | Method | AUROC↑ | AUPR↑ | FPR95↓ |
|---|---|---|---|---|
| Prob-Based | OE | 89.91 | 87.32 | 34.06 |
|  | **+Ours** | **91.52** | **89.03** | **30.64** |
| Energy-Based | Energy | 90.27 | 88.73 | 34.42 |
|  | **+Ours** | **91.41** | **90.63** | **31.81** |
| Dist-Based | Maha | 88.26 | 87.94 | 42.74 |
|  | **+Ours** | **89.30** | **88.68** | **39.54** |

Table 6: Robustness to OOD test sets.

| OOD Dataset | Method | AUROC↑ | AUPR↑ | FPR95↓ |
|---|---|---|---|---|
| Far-OOD | PASCL | 96.63 | 98.06 | 12.18 |
|  | **+Ours** | **97.50** | **98.26** | **9.65** |
| Near-OOD | PASCL | 84.43 | 82.99 | 57.27 |
|  | **+Ours** | **86.61** | **85.71** | **55.51** |
| Spurious-OOD | PASCL | 79.00 | 81.83 | 63.57 |
|  | **+Ours** | **83.27** | **84.19** | **57.69** |

**ImOOD is robust to different OOD test sets.** In the preceding sections, we evaluated our method on the CIFAR10-LT benchmark, where the SCOOD test set [56] comprises 6 subsets covering different scenarios. As suggested by Fort et al. [17], the SVHN subset can be viewed as far OOD, and the

CIFAR100 subset can be seen as near OOD (with CIFAR10-LT as ID). According to the detailed results in Tab. 6, our ImOOD brings consistent enhancement against the strong baseline PASCL regardless of the near or far OOD test set. Furthermore, we also report the spurious OOD detection evaluation in Tab. 6. Specifically, we follow Ming et al. [42] to take WaterBird as the imbalanced ID dataset, which also suffers from the imbalance problem (on water birds and land birds), and a subset of Places [59] as the spurious OOD test set (with spurious correlation to background). Results in Tab. 6 also demonstrate our method's robustness in handling spurious OOD problems, with a considerable improvement of 4.3% increase on AUROC and 5.96% decrease on FPR95. The robustness of ImOOD to various OOD testing scenarios is verified.

### 4.4 ImOOD's Inference-time Application

Despite our main focus on training more balanced OOD detectors, we also make some attempts to apply our method during pre-trained models' inference. According to our Theorem 3.2, for an existing OOD detector $P(i|x)$ (*e.g.*, trained with BinDisc), we can calculate the bias term $\beta(x)$ to regulate the vanilla scorer $P(i|x)$ into balanced $P^{bal}(i|x) = \beta(x) \cdot P(i|x)$. However, as $\beta(x) = \sum_y \gamma_y(x) \frac{P(y|x,i)}{P(y)}$, the estimation of $\gamma_y(x)$ presents considerable difficulty without training, but we have also tried some trivial approaches in Tab. 7.

Table 7: Attempts to apply our ImOOD into pre-trained models' inference stages.

| Method | Detector | AUROC↑ | AUPR↑ | FPR95↓ |
|---|---|---|---|---|
| BinDisc | $P(i|x)$ | 90.06 | 88.72 | 33.39 |
| +**Ours** (infer) | $\beta_1(x)P(i|x)$ | 90.34 | 88.45 | 32.10 |
| +**Ours** (infer) | $\hat{\beta}(x)P(i|x)$ | 90.86 | 88.95 | 30.80 |
| +**Ours** (train) | $\beta(x)P(i|x)$ | **92.23** | **91.92** | **29.95** |

First, we simply treat $\gamma_y(x)$ as a constant (*e.g.*, $\gamma_y(x) \equiv \gamma_1 = 1$) for arbitrary input $x$ and class $y$ to calculate the bias term (denoted as $\beta_1(x)$), and the results on CIFAR10-LT benchmark immediately witness a performance improvement (*e.g.*, 0.28% increase on AUROC and 1.29% decrease on FPR95) compared to the baseline OOD detector. However, the improvement is relatively insignificant, and the phenomenon is consistent with our ablation studies in Tab. 4, which demonstrates the importance of learning a *class-dependent* and *input-dependent* $\gamma_y(x)$ during training.

Then, inspired by the statistical results in Fig. 2, we take a further step to use a polynomial (rank=2) to fit the curve between the predicted class $y$ and $\gamma_y(x)$ learned by another model, and apply the coefficients to estimate a *class-dependent* $\hat{\gamma}_y$ for the baseline model (denoted as $\hat{\beta}(x)P(i|x)$). This operation receives further enhancement on OOD detection and gets close to our learned model (*e.g.*, 30.80% *v.s.* 29.95% of FPR95).

In conclusion, our attempts illustrate the potential of applying our method to an existing model without post-training, and we will continue to extend $\hat{\gamma}_y$ to an *input-dependent* version (say $\hat{\gamma}_y(x)$) in our future work.

## 5 Conclusion and Discussion

This paper establishes a statistical framework ImOOD to formulate OOD detection on imbalanced data distribution. Through theoretical analysis, we find there exists a class-aware biased item between balanced and imbalanced OOD detection models. Based on it, our ImOOD provides a unified training-time regularization technique to alleviate the imbalance problem. On three popular imbalanced OOD detection benchmarks, extensive experiments and ablation studies to demonstrate the validity and versatility of our method. We hope our work can inspire new research in this community.

**Limitations.** Following the literature, ImOOD utilizes auxiliary OOD training samples to refine the the ID/OOD decision boundary. However, unforeseen OOD samples in real-world applications could potentially challenge this boundary. To mitigate this issue, integrating online-learning strategies for adaptive decision-making during testing is a promising avenue. We view this as our future work.

## Acknowledgments and Disclosure of Funding

This work was supported in part by the Fundamental Research Funds for the Central Universities, in part by Alibaba Cloud through the Research Intern Program, and in part by Zhejiang Provincial Natural Science Foundation of China under Grant No. LDT23F01013F01.

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

# A Theorem Proofs

## A.1 Proof for Lemma 3.1

From Bayesian Theorem, we have:

$$P(y|\boldsymbol{x}) = \frac{P(\boldsymbol{x}|y) \cdot P(y)}{P(\boldsymbol{x})} \tag{A1}$$

In class-balanced scenarios, the class prior $P^{\text{bal}}(y)$ equals $\frac{1}{K}$, where $K$ is the class number. The balanced classification posterior is given by:

$$P^{\text{bal}}(y|\boldsymbol{x}) = \frac{P^{\text{bal}}(\boldsymbol{x}|y) \cdot \frac{1}{K}}{P^{\text{bal}}(\boldsymbol{x})} \tag{A2}$$

where the marginal probability $P^{\text{bal}}(\boldsymbol{x}) \equiv P(\boldsymbol{x})$ is independent from balanced or imbalanced class distribution. Combining with Eq. (A1) and Eq. (A2), we have:

$$P^{\text{bal}}(y|\boldsymbol{x}) = \frac{P^{\text{bal}}(\boldsymbol{x}|y)}{P(\boldsymbol{x}|y)} \cdot \frac{1}{K} \cdot \frac{P(y|vx)}{P(y)} \triangleq \gamma_y(\boldsymbol{x}) \cdot \frac{P(y|vx)}{P(y)} \tag{A3}$$

where $\gamma_y(\boldsymbol{x}) = \frac{1}{K} \frac{P^{\text{bal}}(\boldsymbol{x}|y)}{P(\boldsymbol{x}|y)}$ captures the likelihood difference for the same sample x between balanced and imbalanced distributions.

## A.2 Discussion for Lemma 3.1 and Statistic Results in Fig. 2

According to Fig. 2 that depicts the statistical distribution on the CIFAR10-LT benchmark, $\gamma_y(\boldsymbol{x})$ is higher for the ID sample from head classes than tail classes. This phenomenon is consistent with $\gamma_y(\boldsymbol{x}) = \frac{1}{K} \frac{P^{\text{bal}}(\boldsymbol{x}|y)}{P(\boldsymbol{x}|y)}$. As the model has seen sufficient training examples for head classes, the data likelihood $P(\boldsymbol{x}|y^h)$ is approaching the balanced situation of $P^{\text{bal}}(\boldsymbol{x}|y)$, the fraction of $\frac{P^{\text{bal}}(\boldsymbol{x}|y)}{P(\boldsymbol{x}|y)}$ is closed to 1. Meanwhile, since a large number of tail samples are not presented during training, the sample space is narrowed down and the data likelihood $P(\boldsymbol{x}|y)$ for tail classes is over-estimated (even higher than $P^{\text{bal}}(\boldsymbol{x}|y)$), leading to a lower fraction of $\frac{P^{\text{bal}}(\boldsymbol{x}|y)}{P(\boldsymbol{x}|y)}$. Therefore, $\gamma_y(\boldsymbol{x}) = \frac{1}{K} \frac{P^{\text{bal}}(\boldsymbol{x}|y)}{P(\boldsymbol{x}|y)}$ presents higher values for head classes than tail classes. The statistical results are well-aligned with the theoretical analysis.

## A.3 Derivation for Eq. (6)

Substituting the sigmoid function $\sigma(z) = \frac{1}{1+e^{-z}}$ into Eq. (5), we have:

$$\frac{1}{1 + e^{-g^{\text{bal}}(\boldsymbol{x})}} = \beta(\boldsymbol{x}) \cdot \frac{1}{1 + e^{-g(\boldsymbol{x})}} \tag{A4}$$

Consequently, we have:

$$e^{-g(\boldsymbol{x})} = \beta(\boldsymbol{x}) \cdot e^{-g^{\text{bal}}(\boldsymbol{x})} + \beta(\boldsymbol{x}) - 1 \tag{A5}$$

And then:

$$\begin{aligned} g(\boldsymbol{x}) &= -\log \beta(\boldsymbol{x}) \cdot e^{-g^{\text{bal}}(\boldsymbol{x})} + \beta(\boldsymbol{x}) - 1 \\ &= -\log e^{-g^{\text{bal}}(\boldsymbol{x})} \left[ 1 + (\beta(\boldsymbol{x}) - 1) \cdot e^{g^{\text{bal}}(\boldsymbol{x})} \right] \\ &= g^{\text{bal}}(\boldsymbol{x}) - \log \left[ (\beta(\boldsymbol{x}) - 1) \cdot e^{g^{\text{bal}}(\boldsymbol{x})} + 1 \right] \end{aligned} \tag{A6}$$

# B  Experimental Settings and Implementations

## B.1  Training Settings

For a fair comparison, we mainly follow PASCL's [51] settings. On CIFAR10/100-LT benchmarks, we train ResNet18 [18] for 200 epochs using Adam optimizer, with a batch size of 256. The initial learning rate is 0.001, which is decayed to 0 using a cosine annealing scheduler. The weight decay is $5 \times 10^{-4}$. On the ImageNet-LT benchmark, we train ResNet50 [18] for 100 epochs with SGD optimizer with the momentum of 0.9. The batch size is 256. The initial learning rate is 0.1, which is decayed by a factor of 10 at epochs 60 and 80. The weight decay is $5 \times 10^{-5}$. During training, each batch contains an equal number of ID and OOD data samples (*i.e.*, 256 ID samples and 256 OOD samples). We use 2 NVIDIA V100-32G GPUs in all our experiments.

For a better performance, one may carefully tune the hyper-parameters to train the models.

## B.2  Implementation Details

In the manuscript, we proposed a generalized statistical framework to formularize and alleviate the imbalanced OOD detection problem. This section provides more details on implementing different OOD detectors into a unified formulation, *e.g.*, a binary ID/OOD classifier [5]:

- For BinDisc [5], we simply append an extra ID/OOD output node to the classifier layer of a standard ResNet model, where ID classifier $f$ and OOD detector $g$ share the same feature extractor. Then we adopt the sigmoid function to convert the logit $g(\boldsymbol{x})$ into the ID/OOD probability $\hat{p}(i|\boldsymbol{x}) = \frac{1}{1+e^{-g(\boldsymbol{x})}}$.

- For Energy [32], following Du et al. [14], we first compute the negative free-energy $E(\boldsymbol{x}; f) = \log \sum_y e^{f_y(\boldsymbol{x})}$, and then attach an extra linear layer to calculate the ID/OOD logit $g(\boldsymbol{x}) = w \cdot E(\boldsymbol{x}; f) + b$, where $w, b$ are learnable scalars. Hence, the sigmoid function is able to convert the logit $g(\boldsymbol{x})$ into the probability $\hat{p}(i|\boldsymbol{x})$.

- For OE [20], similarly, we compute the maximum softmax-probability $\text{MSP}(\boldsymbol{x}; f) = \max_y \frac{e^{f_y(\boldsymbol{x})}}{\sum_{y'} e^{f_{y'}(\boldsymbol{x})}}$, and use another linear layer to obtain the ID/OOD logit $g(\boldsymbol{x}) = w \cdot \text{MSP}(\boldsymbol{x}; f) + b$.

- For Mahalanobis distance [24], we maintain an online feature-pool for each ID class $y$, so as to calculate the Mahalanobis distances for the test samples as $D_m(\boldsymbol{x})$. Then, an extra linear layer is adopted to transform the distances to ID/OOD logits as $g(\boldsymbol{x}) = w \cdot D_m(\boldsymbol{x}) + b$.

By doing so, one can exploit the unified training-time regularization in Sec. 3.3 to derive a strong OOD detector.

# C  Additional Experimental Results

## C.1  Additional Error Statistics

In this section, we present the class-aware error statistics for OOD detection on different benchmarks (see Fig. A1) and different detectors (see Fig. A2). For each OOD detector on each benchmark, we first compute the OOD scores $g(\boldsymbol{x})$ for all the ID and OOD test data. Then, a threshold $\lambda$ is determined to ensure a high fraction of OOD data (*i.e.*, 95%) is correctly detected as out-of-distribution. Recall that $g(\boldsymbol{x})$ indicates the in-distribution probability for a given sample (*i.e.*, $P(i|\boldsymbol{x})$). Finally, we statistic the distributions of wrongly detected ID/OOD samples.

Specifically, in Fig. A1 and Fig. A2, the first row displays class labels of ID samples that are wrongly detected as OOD (*i.e.*, $g(\boldsymbol{x}) < \lambda$), and the second row exhibits class predictions of OOD samples that are wrongly detected as ID (*i.e.*, $g(\boldsymbol{x}) > \lambda$). In each subplot, we statistic the distribution over *head*, *middle*, and *tail* classes (the division rule follows Wang et al. [51]) for simplicity. Note that the total count of wrong OOD samples (in the second row) is constant, and a better OOD detector $g$ will receive fewer wrong ID samples (in the first row).

Fig. A1 compares our method with PASCL [51] on CIFAR10/100-LT and ImageNet-LT benchmarks. The results indicate our method (dashed bar) performs relatively more balanced OOD detection on

all benchmarks. We reduce the error number of ID samples from tail classes, and simultaneously decrease the error number of OOD samples that are predicted as head classes. In particular, our method achieves considerable balanced OOD detection on ImageNet-LT (the right column), which brings a significant performance improvement, as discussed in Sec. 4.2.

Fig. A2 compares our integrated version and vanilla OOD detectors (*i.e.*, OE, Energy, and BinDisc) on the CIFAR10-LT benchmark. Similarly, the results indicate our method performs relatively more balanced OOD detection against all original detectors. The versatility of our ImOOD framework is validated, as discussed in Sec. 4.3.

However, the statistics in Fig. A1 and Fig. A2 indicate the imbalanced problem has not been fully solved, since the scarce training samples of tail classes still affect the data-driven learning process. More data-level re-balancing techniques may be leveraged to further address this issue.

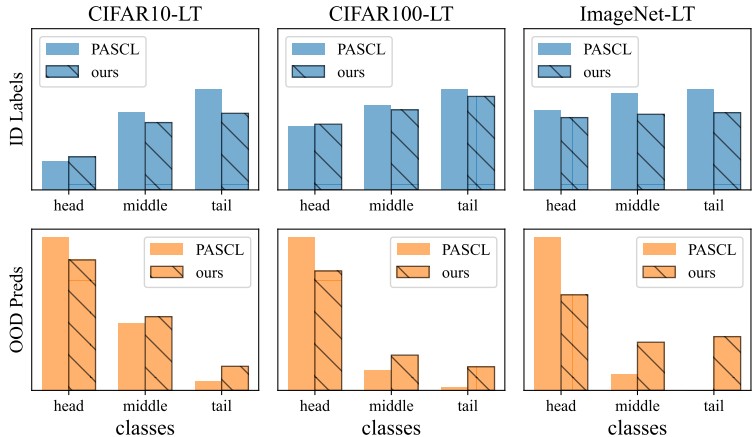

Figure A1: Class-aware *error* statistics for OOD detection on different benchmarks.

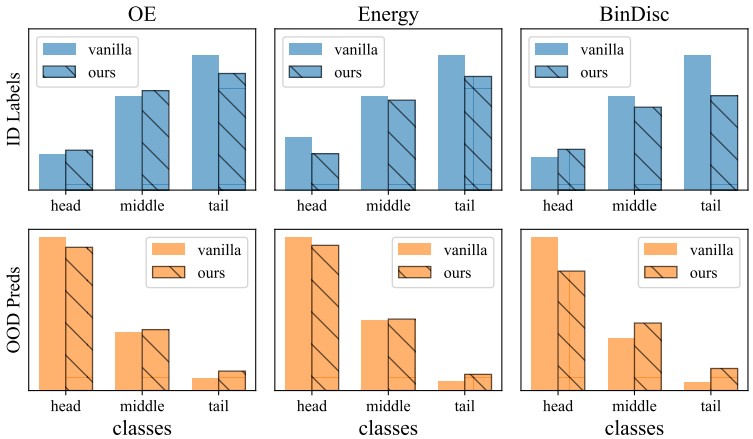

Figure A2: Class-aware *error* statistics for different OOD detectors on CIFAR10-LT.

## C.2 Detailed Results on CIFAR10/100-LT Benchmarks

For CIFAR10/100-LT benchmark, Textures [11], SVHN [43], CIFAR100/10 (respectively), TinyImageNet [27], LSUN [58], and Places365 [59] from SC-OOD dataset [56] are adopted as OOD test sets. The mean results on the six OOD sets are reported in Sec. 4.

In this section, we reported the detailed measures on those OOD test sets in Tab. A1 and Tab. A2, as the supplementary to Tab. 1. The results indicate our method consistently outperforms the state-of-the-art PASCL [51] on most of the subsets.

Table A1: Detailed results on CIFAR10-LT.

| $\mathcal{D}_{out}^{test}$ | Method | AUROC ($\uparrow$) | AUPR ($\uparrow$) | FPR95 ($\downarrow$) |
|---|---|---|---|---|
| Texture | PASCL | $93.16 \pm 0.37$ | $84.80 \pm 1.50$ | $23.26 \pm 0.91$ |
| | **Ours** | $\mathbf{96.58} \pm 0.21$ | $\mathbf{94.09} \pm 0.29$ | $\mathbf{16.22} \pm 0.47$ |
| SVHN | PASCL | $96.63 \pm 0.90$ | $98.06 \pm 0.56$ | $12.18 \pm 3.33$ |
| | **Ours** | $\mathbf{97.50} \pm 0.47$ | $\mathbf{98.26} \pm 0.41$ | $\mathbf{9.65} \pm 0.72$ |
| CIFAR100 | PASCL | $84.43 \pm 0.23$ | $82.99 \pm 0.48$ | $57.27 \pm 0.88$ |
| | **Ours** | $\mathbf{86.61} \pm 0.19$ | $\mathbf{85.71} \pm 0.12$ | $\mathbf{55.51} \pm 0.41$ |
| TIN | PASCL | $87.14 \pm 0.18$ | $81.54 \pm 0.38$ | $47.69 \pm 0.59$ |
| | **Ours** | $\mathbf{88.75} \pm 0.35$ | $\mathbf{86.27} \pm 0.39$ | $\mathbf{40.52} \pm 0.32$ |
| LSUN | PASCL | $93.17 \pm 0.15$ | $91.76 \pm 0.53$ | $26.40 \pm 1.00$ |
| | **Ours** | $\mathbf{94.55} \pm 0.23$ | $\mathbf{93.70} \pm 0.34$ | $\mathbf{22.02} \pm 0.37$ |
| Places365 | PASCL | $91.43 \pm 0.17$ | $96.28 \pm 0.14$ | $33.40 \pm 0.88$ |
| | **Ours** | $\mathbf{93.57} \pm 0.14$ | $\mathbf{97.04} \pm 0.21$ | $\mathbf{28.43} \pm 0.61$ |
| **Average** | PASCL | $90.99 \pm 0.19$ | $89.24 \pm 0.34$ | $33.36 \pm 0.79$ |
| | **Ours** | $\mathbf{92.93} \pm 0.26$ | $\mathbf{92.51} \pm 0.29$ | $\mathbf{28.73} \pm 0.48$ |

Table A2: Detailed results on CIFAR100-LT.

| $\mathcal{D}_{out}^{test}$ | Method | AUROC ($\uparrow$) | AUPR ($\uparrow$) | FPR95 ($\downarrow$) |
|---|---|---|---|---|
| Texture | PASCL | $76.01 \pm 0.66$ | $58.12 \pm 1.06$ | $\mathbf{67.43} \pm 1.93$ |
| | **Ours** | $\mathbf{77.34} \pm 0.55$ | $\mathbf{62.76} \pm 0.69$ | $68.87 \pm 0.44$ |
| SVHN | PASCL | $80.19 \pm 2.19$ | $88.49 \pm 1.59$ | $53.45 \pm 3.60$ |
| | **Ours** | $\mathbf{84.20} \pm 0.34$ | $\mathbf{91.86} \pm 0.50$ | $\mathbf{47.50} \pm 0.32$ |
| CIFAR10 | PASCL | $\mathbf{62.33} \pm 0.38$ | $\mathbf{57.14} \pm 0.20$ | $79.55 \pm 0.84$ |
| | **Ours** | $61.53 \pm 0.43$ | $56.56 \pm 0.42$ | $\mathbf{79.19} \pm 0.65$ |
| TIN | PASCL | $68.20 \pm 0.37$ | $51.53 \pm 0.42$ | $76.11 \pm 0.80$ |
| | **Ours** | $\mathbf{68.42} \pm 0.27$ | $\mathbf{52.15} \pm 0.21$ | $\mathbf{75.54} \pm 0.60$ |
| LSUN | PASCL | $77.19 \pm 0.44$ | $61.27 \pm 0.72$ | $63.31 \pm 0.87$ |
| | **Ours** | $\mathbf{77.68} \pm 0.29$ | $\mathbf{61.66} \pm 0.38$ | $\mathbf{60.32} \pm 0.32$ |
| Places365 | PASCL | $76.02 \pm 0.21$ | $86.52 \pm 0.29$ | $64.81 \pm 0.27$ |
| | **Ours** | $\mathbf{76.19} \pm 0.13$ | $\mathbf{86.79} \pm 0.20$ | $\mathbf{62.48} \pm 0.45$ |
| **Average** | PASCL | $73.32 \pm 0.32$ | $67.18 \pm 0.10$ | $67.44 \pm 0.58$ |
| | **Ours** | $\mathbf{74.21} \pm 0.35$ | $\mathbf{68.60} \pm 0.43$ | $\mathbf{65.65} \pm 0.26$ |

## C.3 Comparison on Different Imbalance Ratios

In our manuscript, we mainly take the default imbalance ratio ($\rho = 100$), which means the least frequent (tail) class only has $\frac{1}{100}$ of training samples than the most frequent (head) class). Here, we follow PASCL [51] to investigate another imbalance ratio of $\rho = 50$ (relatively more balanced than $\rho = 100$) on CIFAR10-LT. According to Tab. A3, our method consistently surpasses PASCL on various imbalance levels, and gains a larger enhancement (*e.g.*, near 2.0% of AUROC) on the more imbalanced scenario with $\rho = 100$, which further demonstrates our effectiveness in mitigating imbalanced OOD detection.

Table A3: Evaluation on different imbalance ratio $\rho$ for the CIFAR10-LT benchmark.

| Method | $\rho = 100$ | | | $\rho = 50$ | | |
|---|---|---|---|---|---|---|
| | AUROC$\uparrow$ | AUPR$\uparrow$ | FPR95$\downarrow$ | AUROC$\uparrow$ | AUPR$\uparrow$ | FPR95$\downarrow$ |
| OE | 89.77 | 87.25 | 34.65 | 93.13 | 91.06 | 24.73 |
| PASCL | 90.99 | 89.24 | 33.36 | 93.94 | 92.79 | 22.08 |
| **Ours** | **92.93** | **92.51** | **28.73** | **94.37** | **94.24** | **19.72** |

## C.4 Investigation on SOTA General OOD Detection Methods

Besides the method specifically designed for imbalanced OOD detection, we also compare with several recently published detectors that aims at general (or, balanced) OOD detection:

- NECO [2] studies the neural collapse phenomenon develops a novel post-hoc method to leverage geometric properties and principal component spaces to identify OOD data.
- fDBD [31] investigates model's decision boundaries and proposes to detect OOD using the feature distance to decision boundaries.
- IDLabel [15] theoretically delineates the impact of ID labels on OOD detection, and utilizes ID labels to enhance OOD detection via representation characterizations through spectral decomposition on the graph.
- ID-like [3] leverages the powerful vision-language model CLIP to identify challenging OOD samples/categories from the vicinity space of the ID classes to further facilitate OOD detection.

The experimental results are displayed in Tab. A4. On the CIFAR10-LT benchmark, our method surpasses the recent OOD detectors by a large margin. For imbalanced data distribution, IDLabel [15] struggles to capture the distributional difference across ID classes, while the pure post-hoc methods NECO [2] and fDBD [31] even get worse results because they fail to generalize to the scenario with highly skewed feature spaces and decision boundaries. On the ImageNet-LT benchmark, even the

incorporation of the powerful CLIP model cannot well address the imbalance problem for ID-like [3], while our method can specifically and effectively facilitate imbalanced OOD detection with the help of our theoretical groundness.

Table A4: Comparison with SOTA general OOD detectors.

| Benchmark | Method | Pub&Year | AUROC↑ | AUPR↑ | FPR95↓ |
|---|---|---|---|---|---|
| CIFAR10-LT | NECO | ICLR'24 | 85.15 | 82.39 | 40.44 |
| | fDBD | ICML'24 | 87.90 | 83.07 | 41.98 |
| | IDLabel | ICML'24 | 90.06 | 88.29 | 34.66 |
| | **Ours** | - | **93.55** | **92.83** | **28.52** |
| ImageNet-LT | ID-like | CVPR'24 | 72.05 | 71.37 | 78.36 |
| | **Ours** | - | **75.84** | **73.19** | **74.96** |

## C.5 Additional Analysis on Correctly-detected ID and OOD Samples

In Fig. 1a, we reveal that the class labels for *wrongly*-identified ID samples and the class predictions for *wrongly*-detected OOD samples are both *sensitive* the to the ID class distribution prior. Here, Fig. A3 indicates the *correctly*-detected ID and OOD samples are *insensitive* to the class distribution.

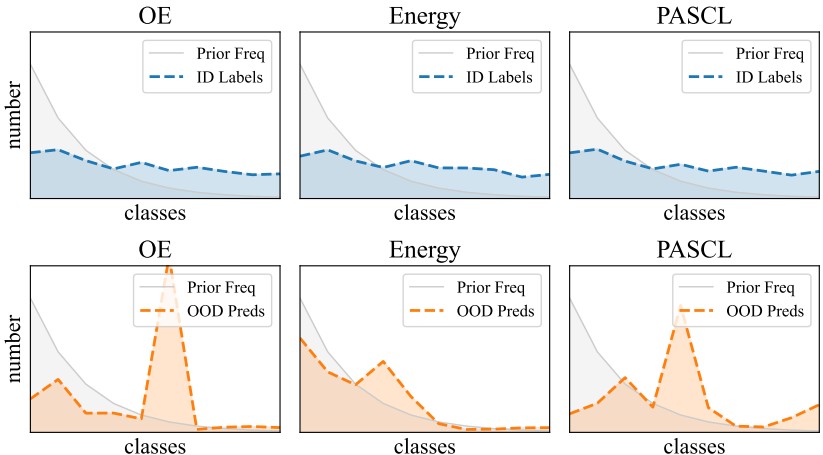

Figure A3: Statistics on *correctly-detected* ID and OOD samples.

In particular, the *per-class prediction quantity* of correctly-detected OOD samples in Fig. A3 may not precisely describe the statistical distribution, as the maximum *ID-class prediction probability* is relatively low (*e.g.*, $max_y P(y|x, i) = 0.12$ for 10-category classification) and introduce extra noise. Therefore, we supplement the statistics of *per-class prediction probability* in Tab. A5, and all of the distributions for OE, Energy, and PASCL are nearly even.

Table A5: Average per-class prediction probability for correctly-detected OOD samples.

| Method | $cls_1$ | $cls_2$ | $cls_3$ | $cls_4$ | $cls_5$ | $cls_6$ | $cls_7$ | $cls_8$ | $cls_9$ | $cls_{10}$ |
|---|---|---|---|---|---|---|---|---|---|---|
| OE | 0.12 | 0.11 | 0.14 | 0.14 | 0.13 | 0.11 | 0.16 | 0.11 | 0.11 | 0.12 |
| Energy | 0.28 | 0.25 | 0.31 | 0.26 | 0.29 | 0.29 | 0.39 | 0.32 | 0.36 | 0.33 |
| PASCL | 0.15 | 0.12 | 0.12 | 0.14 | 0.11 | 0.12 | 0.13 | 0.12 | 0.11 | 0.11 |

