# OpenReview forum: "Rethinking Out-of-Distribution Detection on Imbalanced Data Distribution"
_NeurIPS.cc/2024/Conference — NeurIPS 2024 poster_

### Official Review · Reviewer_XZSm · 2024-07-03

**Soundness:** 3
**Presentation:** 3
**Contribution:** 2
**Rating:** 5
**Confidence:** 3

**Summary:**

The paper proposes the ImOOD framework to address class imbalance issues in OOD detection. Through Bayesian analysis, the authors identify critical biases and provide a unified regularization technique to improve detection performance. They conduct extensive experiments, demonstrating ImOOD's effectiveness on CIFAR10-LT, CIFAR100-LT, and ImageNet-LT datasets.

**Strengths:**

- The paper is well-written and easy to follow.
- The paper offers a solid theoretical foundation, explaining the class-aware bias and providing a unified regularization approach.
- Extensive experiments demonstrate the effectiveness of the proposed method.

**Weaknesses:**

- The method relies on external OOD data, which is often difficult to obtain in OOD detection settings.
- The comparisons in CIFAR and ImageNet experiments seem inconsistent, with fewer methods evaluated on ImageNet. Notably, ClassPrior seems to perform better on ImageNet.
- Some non-long-tail-specific OOD detection methods in the ClassPrior paper also perform well. Given the emergence of many excellent OOD detection methods since ClassPrior's publication, comparing and discussing these methods with the proposed approach would provide a more comprehensive evaluation.

**Questions:**

- Why do the ID labels in Figure 1a show similar fluctuations across OE, Energy, and PASCL?
- I believe the OOD prediction results in Figure 1a do not support the authors' claim. If I am mistaken, please correct me. The distribution of OOD predictions appears consistent with the prior distribution, with no significant deviation in the head region. For OOD samples, since the model hasn't been trained on them, it would predict according to the training sample distribution (prior distribution). I don't see this behavior as problematic.
- Other questions: Please refer to the weaknesses section above.

**Limitations:**

Yes.

---

> ### Author Rebuttal · Authors · 2024-08-07
>
> Dear reviewer XZSm:
>
> We thank the reviewer for the valuable time and constructive suggestions, and our point-to-point responses are presented below:
>
> > **W1**: The method relies on external OOD data, which is often difficult to obtain in OOD detection settings.
>
> **A**: We follow PASCL[1]'s setting to evaluate our method for imbalanced OOD detection with auxiliary OOD data during training, which is a commonly-used setting in the literature[2][3][4] because auxiliary OOD data is relatively easy to obtain in practice[4][5].
>
> On the other hand, to further evaluate our method without real OOD training data, we also integrate with feature-level synthesis methods (e.g., VOS[6] and NPOS[7]) to generate pseudo-OOD data for training.
> In particular, we follow ClassPrior's setting to train a MobileNet model on ImageNet-LT-a8 with VOS for OOD synthesis, and evaluate the performance on four OOD test sets.
> Our method outperforms the SOTA ClassPrior method by a large margin on all subsets.
>
> |Method|iNaturalist||SUN||Places||Textures||**Average**||
> |:--|:--:|:--:|:--:|:--:|:--:|:--:|:--:|:--:|:--:|:--:|
> ||AUROC|FPR95|AUROC|FPR95|AUROC|FPR95|AUROC|FPR95|AUROC|FPR95|
> |ODIN|64.68|93.78|74.29|79.42|69.94|89.70|69.06|82.23|69.49|86.28|
> |GradNorm|70.87|78.12|69.70|67.59|66.00|85.75|63.09|74.89|67.41|76.59|
> |Dice|65.61|86.40|69.35|66.38|65.95|88.42|68.85|68.19|67.44|77.35|
> |ClassPrior|82.51|66.06|80.08|69.12|74.33|79.41|69.58|78.07|76.63|73.16|
> |**Ours**|**86.15**|**59.13**|**81.29**|**65.88**|**77.57**|**76.26**|**72.82**|**72.73**|**79.45**|**68.50**|
>
> > **W2**: The comparisons in CIFAR and ImageNet experiments seem inconsistent, with fewer methods evaluated on ImageNet. Notably, ClassPrior seems to perform better on ImageNet.
>
> **A**: Thanks for pointing out this. The reason is that ClassPrior uses a totally different setting against the literature[1-4] that we follow, and the differences are detailed as follows:
>
> |Settings|Model|ID Data|OOD Training Data|OOD Test Data|
> |:--|:--|:--|:--|:--|
> |ClassPrior|ResNet101/MobileNet|ImageNet-LT-a8|-|iNaturalist & SUN & Places & Textures|
> |Others|ResNet50|ImageNet-LT|ImageNet-1k-Extra|ImageNet-1k-OOD|
>
> In Table 1 of our manuscript, we take ClassPrior's results on CIFAR10/100-LT from COCL's paper. Here, we provide a further comparison on ImageNet following ClassPrior's setting, and the results are reported in the above table (in the response to W1), which demonstrates our superiority against ClassPrior and other OOD detection methods.
>
> > **W3**: Some non-long-tail-specific OOD detection methods in the ClassPrior paper also perform well. Comparing and discussing these methods with the proposed approach would provide a more comprehensive evaluation.
>
> **A**: Thanks for this suggestion. On one hand, after aligning with ClassPrior's setting, we have further illustrated our method's efficacy over other OOD methods.
>
> On the other hand, we have also compared it with several SOTA OOD detection methods published in the most recent top conferences (ICLR'24, CVPR'24, and ICML'24).
> Due to the space limit, please kindly refer to our response to Reviewer QTK3 (R3)'s Question 2 (Q2).
>
> We will add those experiments and discussions to further illustrate the novelty and contribution of our paper.
>
> > **Q1**: Why do the ID labels in Figure 1a show similar fluctuations across OE, Energy, and PASCL?
>
> **A**: In Figure 1a, the `ID labels` actually means `label distribution for **wrongly-detected** ID samples`, and OE, Energy, and PASCL show similar fluctuations because they face the same problem: wrongly detecting samples from tail ID classes as OOD data, which inspires us to formulate this phenomenon and develop a unified solution.
>
> > **Q2**: I believe the OOD prediction results in Figure 1a do not support the authors' claim. If I am mistaken, please correct me. The distribution of OOD predictions appears consistent with the prior distribution, with no significant deviation in the head region. For OOD samples, since the model hasn't been trained on them, it would predict according to the training sample distribution (prior distribution). I don't see this behavior as problematic.
>
> **A**: The fact that models tend to predict according to training data distribution is exactly the core problem for imbalanced OOD detection.
>
> First, we may kindly clarify that Figure 1a displays the class predictions for OOD samples that are **wrongly-detected** as ID data, and all three OOD detectors tend to wrongly detect OOD data as head ID classes. On the other hand, we have shown in our _uploaded PDF file_ that both of the **correctly-detected** ID and OOD samples are **insensitive** to the class distribution.
>
> Therefore, we argue that this behavior (the model will predict according to training distribution) is exactly the problem to solve (e.g., _wrongly_ detecting OOD samples as ID data from head class), which is similar to the problem studied by the long-tailed recognition community (wrongly recognizing test samples as head classes).
> We are thus motivated to formulate the distribution gap on imbalanced OOD detection, and try to regulate the imbalanced detectors towards balanced models using proposed techniques.
>
> We hope our responses can address the reviewer's concerns, and we are more than happy to provide further explanations if there are additional questions.
>
> Best regards,
>
> Authors
>
> ---
>
> [1] Wang et al. Partial and asymmetric contrastive learning for out-of-distribution detection in long-tailed recognition. ICML'22.
>
> [2] Cui et al. Class-balanced loss based on effective number of samples. CVPR'19.
>
> [3] Kang et al. Decoupling representation and classifier for long-tailed recognition. ICLR'20.
>
> [4] Menon et al. Long-tail learning via logit adjustment. ICLR'21.
>
> [5] Yang et al. Generalized out-of-distribution detection: A survey. IJCV'24.
>
> [6] Du et al. Vos: Learning what you don't know by virtual outlier synthesis. ICLR'22.
>
> [7] Tao et al. Non-parametric outlier synthesis. ICLR'23.

---

> ### Comment · Area_Chair_Kybg · 2024-08-12
>
> Dear Reviewers,
>
> Thank you for taking the time to review the paper. The discussion has begun, and active participation is highly appreciated and recommended.
>
> Thanks for your continued efforts and contributions to NeurIPS 2024.
>
> Best regards,
>
> Your Area Chair

---

> ### Author Response · Authors · 2024-08-13
> **Kind Remind of the Ending of Discussion Period**
>
> Dear Reviewer XZSm,
>
> We thank you again for your time and efforts in evaluating our paper. As the discussion period will **end in 24 hours**, we are eager to receive your feedback on our responses, which is very important to us. If there are any further questions, we are more than happy to provide more clarifications as needed.
>
> ---
>
> Best regards,
>
> Authors

---

> > ### Comment · Reviewer_XZSm · 2024-08-14
> >
> > Thank you for your response. (W1) While there are many cases in the literature that rely on OOD data, OOD detection, by definition, assumes the OOD distribution is unknown, so it's important not to make assumptions about it. However, this is not a critical weakness. The responses to w2 and w3 were helpful, thank you.
> >
> > In the rebuttal PDF, the OOD prediction distribution in Figure 1(b) also appears odd. In both OE and PASCAL, there is a large distribution centered on a single class, whereas in Energy, the distribution seems more aligned with the class priors. The authors mentioned in their comments that "both the correctly-detected ID and OOD samples are insensitive to the class distribution," but this doesn't align with my observations. I see two potential issues: First, the OOD prediction distribution in OE and PASCAL might suggest that the OOD sample distribution itself is uneven. Second, the results in Energy do not seem consistent with the authors' conclusions.
> >
> > I have also read all the reviews and your rebuttal. My main concern is about the contribution that IMOOD brings to the community, as the improvements in the experimental results are not particularly impressive. Additionally, I still have some questions about the visualizations. Specifically, the curves in the figures seem unusually erratic and not very smooth, which I find puzzling. I raised this concern in my initial review, but the authors did not address it directly. I suspect this might be related to the limited number of classes.
> >
> > Overall, thank you for your responses. I will adjust my rating to 5, as the concerns outlined above prevent me from giving a higher rating at this time.

---

> > > ### Author Response · Authors · 2024-08-14
> > > **Thanks for Your Timely Feedback**
> > >
> > > Dear Reviewer XZSm,
> > >
> > > We sincerely thank you for your timely feedback and for adjusting your rating.   Here are our further explanations for your new questions.
> > >
> > > Firstly, we are grateful to clarify that the statistics in our original paper are consistent with our motivation.   As for the OOD prediction distribution in Fig 1(b) in our rebuttal PDF file, it describes the _per-class prediction quantity_ of correctly identified OOD samples. In fact, when an OOD sample is identified as OOD data, its maximum ID _prediction probability_ is relatively low (i.e., $max_y{P(y|x,i)} = 0.12$ for 10-category classification), and the quantity statistics on class prediction may thus be biased and unable to capture the true picture.  When we statistic the _per-class prediction probability_, all of the distributions for OE, Energy, and PASCL are nearly even:
> > >
> > > |        | class1 | class2 | class3 | class4 | class5 | class6 | class7 | class8 | class9 | class10 |
> > > |:------:|:------:|:------:|:------:|:------:|:------:|:------:|:------:|:------:|:------:|:-------:|
> > > |   OE   |  0.12  |  0.11  |  0.14  |  0.14  |  0.13  |  0.11  |  0.16  |  0.11  |  0.11  |   0.12  |
> > > | Energy |  0.28  |  0.25  |  0.31  |  0.26  |  0.29  |  0.29  |  0.39  |  0.32  |  0.36  |   0.33  |
> > > |  PASCL |  0.15  |  0.12  |  0.12  |  0.14  |  0.11  |  0.12  |  0.13  |  0.12  |  0.11  |   0.11  |
> > >
> > > Therefore, we claim _both the correctly-detected ID and OOD samples are insensitive to the class distribution_.   We will clarify this in our revised manuscript.
> > >
> > > Secondly, regarding the contribution of our ImOOD paper, we believe our theoretically grounded analysis framework and consistent improvements across datasets and metrics can support our novelty and bring insights to the community.
> > >
> > > Finally, for the curves in the figures, we also statistic the distribution on the ImageNet benchmark with 1,000 categories, which presents similar patterns in the previous analysis.   As the rebuttal PDF cannot be updated during this period, we will include it in our final version.
> > >
> > > We thank you once again for evaluating our paper, and we hope our responses can help address your new concerns.   If there are any questions left, we are glad to provide more explanations.
> > >
> > > ---
> > >
> > > Best regards,
> > >
> > > Authors

---

### Official Review · Reviewer_QTK3 · 2024-07-10

**Soundness:** 2
**Presentation:** 2
**Contribution:** 2
**Rating:** 5
**Confidence:** 4

**Summary:**

This manuscript introduces ImOOD, a statistical framework addressing the OOD detection problem in imbalanced data distributions, identifying common issues such as misidentifying tail class ID samples and erroneously predicting OOD samples as head class ID. It reveals a class-aware bias between balanced and imbalanced OOD detection and proposes a unified training-time regularization technique to mitigate this bias. The method demonstrates consistent performance improvements on benchmarks such as CIFAR10-LT, CIFAR100-LT, and ImageNet-LT, surpassing several state-of-the-art OOD detection methods.

**Strengths:**

1. Theoretical Analysis and Insight: The manuscript introduces a theoretical analysis that reveals a class-aware bias between balanced and imbalanced OOD detection, providing a deeper understanding of the underlying challenges in OOD detection tasks.

2. Strong Experimental Results: The experimental results showcase the method's effectiveness and scalability, demonstrating consistent improvements across various benchmarks, which highlights the robustness and practical applicability of the proposed solution.

**Weaknesses:**

Although the manuscript provides insights through theoretical analysis and proposes a method to improve out-of-distribution (OOD) detection in class-imbalanced scenarios, it essentially estimates a class-specific scaling factor, which is a core aspect of long-tailed recognition and class-imbalanced learning. This is neither a new nor an interesting problem. Additionally, the proposed parametric mapping increases the difficulty of network optimization. It seems that OOD detection under class imbalance is merely an overlap of the two tasks: class-imbalanced learning and OOD detection. Using techniques from both fields can address this issue quite well. Therefore, the significance of the problem addressed in the manuscript is questionable and warrants further consideration.

**Questions:**

Why not obtain the class-specific scaling factor using methods from the long-tailed image recognition domain? Additionally, there is a lack of sufficient experimental evidence to show that using parametric mapping to obtain the class-specific scaling factor is more effective. The paper compares relatively few OOD detection methods; it is recommended to include more recent OOD detection approaches. The manuscript also lacks a summary of related works in class-imbalanced learning.

**Limitations:**

Please see Weaknesses and Questions.

---

> ### Author Rebuttal · Authors · 2024-08-07
>
> Dear reviewer QTK3:
>
> We thank the reviewer for the valuable time and constructive suggestions, and our point-to-point responses are presented below:
>
> > **W1**: It seems that OOD detection under class imbalance is merely an overlap of the two tasks: class-imbalanced learning and OOD detection. Using techniques from both fields can address this issue quite well.
>
> **A**: OOD detection and class-imbalanced learning are _not_ seperate tasks. This issue has already been investigated and verified by PASCL[1]. According to the results on the CIFAR10-LT benchmark below, a simple combination of popular long-tailed learning methods (e.g., Reweighting[2], $\tau$-norm[3], LA[4], etc.) does not address well the imbalanced OOD detection problem. Instead, PASCL treats imbalanced ID classification and OOD detection as a **joint** problem and achieves considerable improvement over the baselines, and our method further boosts imbalanced OOD detection significantly with the help of our theoretical groundedness. We hope the discussion and experiments can help establish the significance and contribution of our paper.
>
> |OOD + LTR|AUROC|AUPR|FPR95|
> |-|:--:|:--:|:--:|
> |OE + None|89.92|87.71|34.80|
> |OE + Reweight|89.34|86.39|37.09|
> |OE + $\tau$-norm|89.58|85.88|33.80|
> |OE + LA|89.46|86.39|34.94|
> |PASCL|90.99|89.24|33.36|
> |**Ours**|**92.93**|**92.51**|**28.73**|
>
> > **Q1**: Why not obtain the class-specific scaling factor using methods from the long-tailed image recognition domain?
>
> **A**: In our paper, the estimate of class prior $P(y)$ is actually borrowed from the long-tailed recognition literature[4] (using label frequency), while we have not currently[5] seen a reliable method to calculate the class-specific factor over balanced and imbalanced data likelihood $\gamma_y(x) = \frac{1}{K}\frac{P^{bal}(x|y)}{P(x|y)}$. Therefore, we choose to learn it by automatic parametric mapping, which has been proven effective[6] when estimating the data density $P(x)$, and the empirical and statistical results in Table 3 and Figure 2 have further confirmed the effectiveness of our estimated $\gamma_y(x)$. Despite that, we are also glad to integrate our method into reliable estimates of $\gamma_y(x)$ for a further study if applicable.
>
>
> > **Q2**: The paper compares relatively few OOD detection methods; it is recommended to include more recent OOD detection approaches.
>
> **A**: Thanks for this suggestion. We have additionally compared with several SOTA OOD detection methods recently published in top conferences (ICLR'24, CVPR'24, and ICML'24):
>
> * NECO[7] studies the neural collapse phenomenon develops a novel post-hoc method to leverage geometric properties and principal component spaces to identify OOD data.
> * fDBD[8] investigates model's decision boundaries and proposes to detect OOD using the feature distance to decision boundaries.
> * IDLabel[9] theoretically delineates the impact of ID labels on OOD detection, and utilizes ID labels to enhance OOD detection via representation characterizations through spectral decomposition on the graph.
> * ID-like[10] leverages the powerful vision-language model CLIP to identify challenging OOD samples/categories from the vicinity space of the ID classes, so as to further facilitate OOD detection.
>
> The experimental results are displayed as follows:
>
> |Benchmark|Method|Pub&Year|AUROC|AUPR|FPR95|
> |:--|:--:|:--:|:--:|:--:|:--:|
> |CIFAR10-LT|NECO|ICLR'24|85.15|82.39|40.44|
> ||fDBD|ICML'24|87.90|83.07|41.98|
> ||IDLabel|ICML'24|90.06|88.29|34.66|
> ||**Ours**|**-**|**93.55**|**92.83**|**28.52**|
> |Imagenet-LT|ID-like|CVPR'24|72.05|71.37|78.36|
> ||**Ours**|**-**|**75.84**|**73.19**|**74.96**|
>
> On the CIFAR10-LT benchmark, our method surpasses the recent OOD detectors by a large margin. For imbalanced data distribution, IDLabel[9] struggles to capture the distributional difference across ID classes, while the pure post-hoc methods NECO[7] and fDBD[8] even get worse results because they fail to generalize to the scenario with highly skewed feature spaces and decision boundaries. On the ImageNet-LT benchmark, even the incorporation of the powerful CLIP model cannot well address the imbalance problem for ID-like[10], while our method can specifically and effectively facilitate imbalanced OOD detection with the help of our theoretical groundness.
>
> We will add those experiments and discussions to further illustrate the novelty and contribution of our paper.
>
> > **Q3**: The manuscript also lacks a summary of related works in class-imbalanced learning.
>
> **A**: Thanks for this suggestion. We will supplement the discussion of current class-imbalanced learning methods[5] (via class re-balancing, information augmentation, module improvement, etc.), and specifically illustrate that the imbalanced ID recognition and OOD detection are a joint problem in the literature (the response to W1) to highlight our contribution.
>
> We hope our responses can address the reviewer's concerns, and we are more than happy to provide further explanations if there are additional questions.
>
> Best regards,
>
> Authors
>
> ---
>
> [1] Wang et al. Partial and asymmetric contrastive learning for out-of-distribution detection in long-tailed recognition. ICML'22.
>
> [2] Cui et al. Class-balanced loss based on effective number of samples. CVPR'19.
>
> [3] Kang et al. Decoupling representation and classifier for long-tailed recognition. ICLR'20.
>
> [4] Menon et al. Long-tail learning via logit adjustment. ICLR'21.
>
> [5] Zhang et al. Deep long-tailed learning: A survey. TPAMI'23.
>
> [6] Kumar et al. Normalizing flow based feature synthesis for outlier-aware object detection. CVPR'23.
>
> [7] Ammar et al. NECO: NEural Collapse Based Out-of-distribution Detection. ICLR'24.
>
> [8] Liu et al. Fast Decision Boundary based Out-of-Distribution Detector. ICML'24.
>
> [9] Du et al. When and how does in-distribution label help out-of-distribution detection? ICML'24.
>
> [10] Bai et al. ID-like Prompt Learning for Few-Shot Out-of-Distribution Detection. CVPR'24.

---

> ### Comment · Area_Chair_Kybg · 2024-08-12
>
> Dear Reviewers,
>
> Thank you for taking the time to review the paper. The discussion has begun, and active participation is highly appreciated and recommended.
>
> Thanks for your continued efforts and contributions to NeurIPS 2024.
>
> Best regards,
>
> Your Area Chair

---

> ### Author Response · Authors · 2024-08-13
> **Kind Remind of the Ending of Discussion Period**
>
> Dear Reviewer QTK3,
>
> We thank you again for your time and efforts in evaluating our paper. As the discussion period will **end in 24 hours**, we are eager to receive your feedback on our responses, which is very important to us. If there are any further questions, we are more than happy to provide more clarifications as needed.
>
> ---
>
> Best regards,
>
> Authors

---

> > ### Comment · Reviewer_QTK3 · 2024-08-13
> >
> > Thank you authors' response. I will increase the score.

---

> ### Author Response · Authors · 2024-08-13
> **Thanks for Reviewer's Positive Feedback!**
>
> Dear Reviewer QTK3,
>
> Thank you for your positive feedback and for raising your score! We are glad that our responses addressed your concerns and will ensure to include them in the final version.
>
> ---
>
> Best regards,
>
> Authors

---

### Official Review · Reviewer_E388 · 2024-07-12

**Soundness:** 3
**Presentation:** 3
**Contribution:** 3
**Rating:** 6
**Confidence:** 3

**Summary:**

The paper addresses the challenge of detecting and rejecting out-of-distribution (OOD) samples by neural networks, particularly when the in-distribution (ID) data is inherently imbalanced. The authors observe that existing OOD detection methods struggle under these conditions primarily because they either misclassify ID samples from minority (tail) classes as OOD or mistakenly identify OOD samples as belonging to majority (head) classes.

To tackle this issue, the authors introduce a new statistical framework called ImOOD, which is designed to understand and improve OOD detection in the presence of imbalanced data. The framework considers the class-aware biases that arise due to data imbalance and incorporates a unified training-time regularization technique aimed at mitigating these biases.

**Strengths:**

The introduction of the ImOOD framework provides a new way to conceptualize and address the problems arising from data imbalance in OOD detection.

The paper not only proposes a theoretical model but also validates it with empirical results, showing consistent performance improvements across multiple datasets.

**Weaknesses:**

A minor issue is that, when we know in advance that the data are long-tailed, it is a common practice that we will use the long-tailed learning methods to counteract its impacts. In this case, is the study of long tailed OOD detection an important topic? Or, even with long-tailed learning, the OOD detection therein still be a critical issue?

I am not sure if the prior information can be properly estimated, considering the situations that the adopted models are not well calibrated (otherwise, OOD detection will not be a challenging task).

**Questions:**

Please refer to the Weaknesses

**Limitations:**

Please refer to the Weaknesses

---

> ### Author Rebuttal · Authors · 2024-08-07
>
> Dear reviewer E388:
>
> We thank the reviewer for the valuable time and constructive suggestions, and our point-to-point responses are presented below:
>
> > **W1**: A minor issue is that, when we know in advance that the data are long-tailed, it is a common practice that we will use the long-tailed learning methods to counteract its impacts. In this case, is the study of long tailed OOD detection an important topic? Or, even with long-tailed learning, the OOD detection therein still be a critical issue?
>
> **A**: Yes, this issue has already been investigated and verified by PASCL[1]. According to the results on the CIFAR10-LT benchmark below, a simple combination of popular long-tailed learning methods (e.g., Reweighting[2], $\tau$-norm[3], LA[4], etc.) does not address well the imbalanced OOD detection problem. Instead, PASCL treats imbalanced ID classification and OOD detection as a **joint** problem and achieves considerable improvement over the baselines, and our method further boosts imbalanced OOD detection significantly with the help of our theoretical groundedness.
>
> |OOD + LTR|AUROC|AUPR|FPR95|
> |:-------------|:---------:|:---------:|:---------:|
> |OE + None|89.92|87.71|34.80|
> |OE + Reweight|89.34|86.39|37.09|
> |OE + $\tau$-norm|89.58|85.88|33.80|
> |OE + LA|89.46|86.39|34.94|
> |PASCL|90.99|89.24|33.36|
> |**Ours**|**92.93**|**92.51**|**28.73**|
>
> > **W2**: I am not sure if the prior information can be properly estimated, considering the situations that the adopted models are not well calibrated (otherwise, OOD detection will not be a challenging task).
>
> **A**: In previous studies, statistics on label frequency is a commonly-adopted estimate for class prior information (i.e., $P(y) := \frac{n_y}{n}$)[4][5], and automatically learning the data density $P(x)$ (e.g., by normalizing flow[6]) is also proved applicable. Therefore, we follow the literature to estimate the class-aware and input-dependent scaling factor $\gamma_y(x) = \frac{1}{K}\frac{P^{bal}(x|y)}{P(x|y)}$ by automatic optimization, and our empirical statistics on Figure 2 has validated the estimated results.
> On the other hand, we also believe that a more precise estimation will further enhance our methods.
>
> We hope our responses can address the reviewer's concerns, and we are more than happy to provide further explanations if there are additional questions.
>
> Best regards,
>
> Authors
>
> ---
>
> [1] Wang et al. Partial and asymmetric contrastive learning for out-of-distribution detection in long-tailed recognition. ICML'22.
>
> [2] Cui et al. Class-balanced loss based on effective number of samples. CVPR'19.
>
> [3] Kang et al. Decoupling representation and classifier for long-tailed recognition. ICLR'20.
>
> [4] Menon et al. Long-tail learning via logit adjustment. ICLR'21.
>
> [5] Jiang et al. Detecting out-of-distribution data through in-distribution class prior. ICML'23.
>
> [6] Kumar et al. Normalizing flow based feature synthesis for outlier-aware object detection. CVPR'23.

---

> ### Author Response · Authors · 2024-08-13
> **Kind Remind of the Ending of Discussion Period**
>
> Dear Reviewer E388,
>
> We thank you again for your time and efforts in evaluating our paper. As the discussion period will **end in 24 hours**, we are eager to receive your feedback on our responses, which is very important to us. If there are any further questions, we are more than happy to provide more clarifications as needed.
>
> ---
>
> Best regards,
>
> Authors

---

### Official Review · Reviewer_5Lex · 2024-07-13

**Soundness:** 3
**Presentation:** 3
**Contribution:** 3
**Rating:** 6
**Confidence:** 3

**Summary:**

The paper focuses on imbalanced data distribution, and finds that there is a bias term between balanced and imbalanced classification which can be used to explain the performance gap. To account for this bias, the authors introduce a regularization term, and show improved results on benchmarks.

**Strengths:**

- It is interesting to quantify the performance gap, and the proposed regularization technique is theoretically grounded and addresses the issue.
- ImOOD can be generalized to existing OOD detection techniques and can also be combined with methods which use auxiliary data, making it easily adaptable.
- The method achieves outperforms other OOD detection schemes across datasets and metrics.
- The ablations presented are compelling and highlight the intended performance of ImOOD.

**Weaknesses:**

- Train-time regularization techniques cannot be applied to existing pretrained models, which may limit their adoption.
- The method requires OOD data during training, and results may not generalize to other unseen OOD settings.

**Questions:**

- Have you tested the benefits of your method with varying levels of imbalance? How does the performance change?

**Limitations:**

See weaknesses.

---

> ### Author Rebuttal · Authors · 2024-08-07
>
> Dear reviewer 5Lex:
>
> We thank the reviewer for the valuable time and constructive suggestions, and our point-to-point responses are presented below:
>
> > **W1**: Train-time regularization techniques cannot be applied to existing pretrained models, which may limit their adoption.
>
> **A**: We have also tried to apply our method during pre-trained models' inference. According to our Theorem 3.2 in our manuscript, for an existing OOD detector $P(i|x)$ (e.g., trained with BinDisc), we can calculate the bias term $\beta(x)$ to regulate the vanilla scorer into balanced $P^{bal}(i|x) = \beta(x) \cdot P(i|x)$. However, as $\beta(x) = \sum_y{\gamma_y(x)\frac{P(y|x,i)}{P(y)}}$, the estimation of $\gamma_y(x)$ presents considerable difficulty without training, but we have also tried some trivial approaches to adapting our method to inference time.
>
> |Method|Detector|AUROC|AUPR|FPR95|
> |-|-|:--:|:--:|:--:|
> |BinDisc|$P(i\|x)$|90.06|88.72|33.39|
> |+Ours (infer)|$\beta_1(x)P(i\|x)$|90.34|88.45|32.10|
> |+Ours (infer)|$\hat{\beta}(x)P(i\|x)$|90.86|88.95|30.80|
> |+Ours (train)|$\beta(x)P(i\|x)$|**92.23**|**91.92**|**29.95**|
>
> First, we simply treat $\gamma_y(x)$ as a constant (e.g., $1$) for arbitrary input $x$ and class $y$ to calculate the bias term (denoted as $\beta_1(x)$), and the results on CIFAR10-LT benchmark immediately witness a performance improvement (e.g., 0.28% increase on AUROC and 1.29% decrease on FPR95) compared to the baseline OOD detector.
> However, the improvement is relatively insignificant, and the phenomenon is consistent with our ablation studies in Table 3, which demonstrates the importance of learning a _class-dependent_ and _input-dependent_ $\gamma_y(x)$ during training.
>
> Then, inspired by the statistical results in Figure 2, we take a further step to use a polynomial (rank=2) to fit the curve between the predicted class $y$ and $\gamma_y(x)$ learned by another model, and apply the coefficients to estimate a _class-dependent_ $\hat{\gamma}_y$ for the baseline model (denoted as $\hat{\beta}(x)P(i|x)$).
> This operation receives further enhancement on OOD detection and gets close to our learned model (e.g., 30.80% v.s. 29.95% of FPR95).
>
> In conclusion, our attempts illustrate the potential of applying our method to an existing model without post-training, and we will continue to extend $\hat{\gamma}_y$ to an _input-dependent_ version (say $\hat{\gamma}_y(x)$) in our future work.
>
>
> > **W2**: The method requires OOD data during training, and results may not generalize to other unseen OOD settings.
>
> **A**: We follow PASCL[1]'s setting to evaluate our method in imbalanced OOD detection with auxiliary OOD data during training, which is a commonly used setting in the literature[2][3][4] because auxiliary OOD data is relatively easy to obtain and usually generalizes well to unseen OOD samples[2][5].
>
> On the other hand, to further evaluate our method without real OOD training data, we also integrate with feature-level synthesis methods (e.g., VOS[6] and NPOS[7]) to generate pseudo-OOD data for training.
> In particular, we follow ClassPrior[8]'s setting to train a MobileNet model on ImageNet-LT-a8 with VOS for OOD synthesis, and evaluate the performance on four OOD test sets.
> Our method outperforms the SOTA ClassPrior method by a large margin on all subsets.
>
> |Method|iNaturalist||SUN||Places||Textures||**Average**||
> |:--|:--:|:--:|:--:|:--:|:--:|:--:|:--:|:--:|:--:|:--:|
> ||AUROC|FPR95|AUROC|FPR95|AUROC|FPR95|AUROC|FPR95|AUROC|FPR95|
> |ClassPrior|82.51|66.06|80.08|69.12|74.33|79.41|69.58|78.07|76.63|73.16|
> |**Ours**|**86.15**|**59.13**|**81.29**|**65.88**|**77.57**|**76.26**|**72.82**|**72.73**|**79.45**|**68.50**|
>
>
> > **Q1**: Have you tested the benefits of your method with varying levels of imbalance? How does the performance change?
>
> **A**: Yes. In our original manuscript, we mainly take the default imbalance ratio ($\rho=100$), which means the least frequent (tail) class only has $\frac{1}{100}$ of training samples than the most frequent (head) class). Here, we follow PASCL to investigate another imbalance ratio of $\rho=50$ (relatively more balanced than $\rho=100$) on CIFAR10-LT, and the results are presented as follows:
>
> |Method|$\rho=100$|||$\rho=50$|||
> |--|:----:|:---:|:----:|:---:|:----:|:-----:|
> ||AUROC|AUPR|FPR95|AUROC|AUPR|FPR95|
> |OE|89.77|87.25|34.65|93.13|91.06|24.73|
> |PASCL|90.99|89.24|33.36|93.94|92.79|22.08|
> |**Ours**|**92.93**|**92.51**|**28.73**|**94.37**|**94.24**|**19.72**|
>
> Accordingly, our method consistently surpasses PASCL on various imbalance levels. Specifically, our method gains a larger enhancement (e.g., near 2.0% of AUROC) on the more imbalanced scenario with $\rho=100$, which further demonstrates our effectiveness in mitigating imbalanced OOD detection.
>
> We hope our responses can address the reviewer's concerns, and we are more than happy to provide further explanations if there are additional questions.
>
> Best regards,
>
> Authors
>
> ---
>
> [1] Wang et al. Partial and asymmetric contrastive learning for out-of-distribution detection in long-tailed recognition. ICML'22.
>
> [2] Hendrycks et al. Deep Anomaly Detection with Outlier Exposure. ICLR'19.
>
> [3] Wei et al. EAT: Towards Long-Tailed Out-of-Distribution Detection. AAAI'24.
>
> [4] Miao et al. Out-of-distribution detection in long-tailed recognition with calibrated outlier class learning. AAAI'24.
>
> [5] Yang et al. Generalized out-of-distribution detection: A survey. IJCV'24.
>
> [6] Du et al. Vos: Learning what you don't know by virtual outlier synthesis. ICLR'22.
>
> [7] Tao et al. Non-parametric outlier synthesis. ICLR'23.
>
> [8] Jiang et al. Detecting out-of-distribution data through in-distribution class prior. ICML'23.

---

> > ### Comment · Reviewer_5Lex · 2024-08-12
> >
> > Thank you for the clarifications and additional experimental results! My questions have been addressed.

---

> > > ### Author Response · Authors · 2024-08-12
> > > **Thanks for your timely feedback**
> > >
> > > Dear Reviewer 5Lex,
> > >
> > > We are sincerely glad that our responses have successfully addressed your concerns. Thanks again for your valuable time in evaluating our work.
> > >
> > > Best regards,
> > >
> > > Authors

---

### Author Rebuttal · Authors · 2024-08-07

We thank all reviewers for their valuable time and constructive suggestions when evaluating our manuscript. We are really encouraged to see **ALL** reviewers find our method **interesting and theoretically grounded** to formulate the gap for imbalanced OOD detection, and **effective and generalized** across datasets and metrics with consistently significant improvements.

We have provided point-to-point responses to reviewers' comments below, and here is a brief summary of the included experiments and explanations:

* **Preliminary investigation/verification on the joint imbalanced OOD detection problem**. We have supplemented the preliminary discussions (originated from PASCL) on the joint problem of imbalanced recognition and OOD detection, so as to enhance the significance and contribution of our research topic.

* **Additional comparison with more general OOD detection approaches**. We have newly provided more comparison and discussion with several SOTA approaches on general OOD detection from recent top conferences, and emphasize the necessity and efficacy of building our imbalanced OOD detection method with theoretical groundness.

* **Further exploration on ClassPrior's setting without auxiliary OOD data**. We have explored the scenario without auxiliary OOD training data after aligning with ClassPrior's experimental settings, so as to better illustrate the effectiveness and applicability of our proposed method.

* **Promising attempts to adapt our methods to models' inference stage**. We have attempted to apply our approach to pre-trained models during the inference stage, and shown promising results and potential applications under proper estimations.

We believe reviewers' comments have made our paper much stronger, and we hope our work can further inspire the community to a deeper study into the imbalanced OOD detection problem.

---

### Comment · Area_Chair_Kybg · 2024-08-08
**The discussion has begun**

Dear Reviewers,

Thank you for taking the time to review the paper. The discussion has begun, and active participation is highly appreciated and recommended.

Thanks for your continued efforts and contributions to NeurIPS 2024.

Best regards,

Your Area Chair

---

### Decision · Program_Chairs · 2024-09-25

**Decision:**

Accept (poster)

**Comment:**

An imbalanced situation will indeed influence the OOD detection performance, which is an important research topic in this area. This paper thoroughly investigated both theoretical and empirical perspectives of this problem. All reviewers appreciate the main contribution of this paper and agree that this paper can have a valuable contribution to this field. The authors should carefully revise their paper to reduce the confusing portion of the paper, based on reviewers' comments. Another note from my side is that the first phenomenon "misidentifying tail class ID samples as OOD, while erroneously predicting OOD samples as a head class from ID" was empirically discovered by Jiang (2023) as well. The authors should carefully shape their contributions in the next version.